# Multi-order Orchestrated Curriculum Distillation for Model-Heterogeneous Federated Graph Learning

**Guancheng Wan**[1†], **Xu Cheng**[1†], **Run Liu**[1], **Wenke Huang**[1], **Zitong Shi**[1],
**Pinyi Jin**[1], **Guibin Zhang**[2], **Bo Du**[1*], **Mang Ye**[1*]
[1]Wuhan University    [2]NUS
{guanchengwan, yemang}@whu.edu.cn

## Abstract

Federated Graph Learning (FGL) has been shown to be particularly effective in enabling collaborative training of Graph Neural Networks (GNNs) in decentralized settings. Model-heterogeneous FGL further enhances practical applicability by accommodating client preferences for diverse model architectures. However, existing model-heterogeneous approaches primarily target Euclidean data and fail to account for a crucial aspect of graph-structured data: topological relationships. To address this limitation, we propose `TRUST`, a novel knowledge distillation-based model-heterogeneous FGL framework. Specifically, we propose Progressive Curriculum Node Scheduler to progressively introduce challenging nodes based on learning difficulty. In Adaptive Curriculum Distillation Modulator, we propose an adaptive temperature modulator that dynamically adjusts knowledge distillation temperature to accommodate varying client capabilities and graph complexity. Moreover, we leverage Wasserstein-Driven Affinity Distillation to enable models to capture cross-class structural relationships through optimal transport. Extensive experiments on multiple graph benchmarks and model-heterogeneous settings show that `TRUST` outperforms existing methods, achieving an average 3.6% ↑ performance gain, particularly under moderate heterogeneity conditions. The code is available for anonymous access at `https://github.com/GuanchengWan/TRUST`.

## 1 Introduction

Federated Learning (FL) [20, 19, 18, 43] has emerged as a distributed machine learning paradigm that enables multiple clients to collaboratively train a global model without sharing their privacy-sensitive data, thereby preserving data confidentiality. Traditional FL methods operate by aggregating locally computed model updates (*e.g.*, gradients or weights) from participating clients under the coordination of a central server, eliminating the need for direct data exchange. A prominent branch of FL is Federated Graph Learning (FGL) [53, 6, 41, 3, 5, 44, 42], which specializes in handling graph-structured data. In addition to inheriting privacy-preserving benefits of FL, FGL usually leverages Graph Neural Networks (GNNs) [21, 11, 45, 30] to capture topology information in graph data, offering a flexible and expressive framework for modeling graph-structured information.

Although many existing FGL works have made significant progress in improving the performance of the global model, these studies are often based on the assumption that client models follow the same architecture. This assumption rarely holds in real-world scenarios, where computational resources and task requirements vary across participants and they prefer to design private models independently rather than agreeing on a unified model architecture [54], ultimately restricting their real-world

---

[†] Equal Contribution.
[*] Corresponding Author.

39th Conference on Neural Information Processing Systems (NeurIPS 2025).

applicability. This challenge is formally termed **model-heterogeneous federated learning**. To address this challenge, recent works have proposed several solutions. For instance, pFedHR [48] generates a personalized model for each client through model reassembly to transfer knowledge between clients. DESA [15] leverages synthetic global data to distill knowledge from other client models. FedTGP [56] employs server-side maintained global prototypes to bridge heterogeneous models. However, these methods are primarily tailored for traditional data types like images and do not generalize to non-Euclidean graph-structured data. This gap motivates our core research question:

*How can we design a model-heterogeneous FL framework specifically tailored for graph data?*

Some previous model-heterogeneous FL methods leverage knowledge distillation (KD) [13, 9, 50, 45] to transfer knowledge between clients [16, 24, 54]. For example, FedType [49] introduces small identical proxy models for clients to bridge the global architecture discrepancy, and then they leverage KD to transfer knowledge between large private and small proxy models. However, in the model-heterogeneous FGL scenario, there are significant differences in model architectures and computational capabilities between clients, and the graph data itself has high-order information such as complex non-Euclidean topology, multi-hop path pattern, and community structure. Traditional knowledge distillation methods maintain a constant "distillation difficulty" (fixed

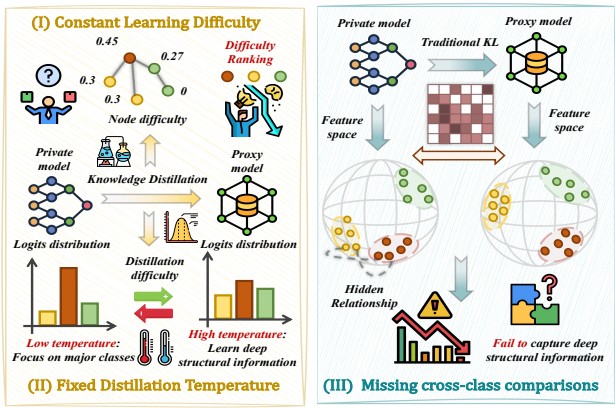

Figure 1: **Problem Illustration**. We describe three challenges **model-heterogeneous FGL** encounters: constant learning difficulty and fixed distillation temperature limit client-specific adaptation and topology preservation. And the lack of cross-class comparison impedes the capture of deep structural graph relationships.

task complexity) [10, 58] throughout the training process. This *one-size-fits-all* approach creates dual dilemmas: For lightweight models, the early introduction of higher-order topology information causes learning bottlenecks and slow convergence. Conversely, for large-scale models with strong expressive ability, it may be too easy for them to exploit their full capacity, resulting in inadequate capture of global structure information. This *fixed-difficulty paradigm* cannot take into account the requirements brought about by model heterogeneity, preventing progressive learning and ultimately leading to degraded FGL Performance. This naturally raises the following question: **I)** *How can we design a KD strategy that dynamically adjusts the task difficulty to accommodate heterogeneous architectures and complex graph topologies?*

In addition to the KD task itself, current KL-divergence based KD approaches maintain static temperature parameters (controlling label distribution smoothness), which govern transfer intensity. This rigidity prevents adaptation to varying graph complexity, label sparsity, and noise patterns. To be specific, KD at low temperatures will neglect small probability events, blurring subtle but important structural information like multi-hop paths and community features. In contrast, high temperatures lose fine-grained node-level information. This inflexibility in distillation signaling hinders client differentiation and deep topological knowledge transfer, ultimately limiting the capacity of the proxy model. This leads us to think that: **II)** *How can we devise an adaptive KD scheme that dynamically calibrates temperature to harmonize global topology transfer with fine-grained node details?* Furthermore, the mentioned KD methods rely on KL-divergence loss, which only compares probability distributions within the same class. This approach lacks cross-class comparison mechanisms, making proxy models fail to capture topological relationships across different categories. This limitation leads to the third problem: **III)** *How can we enable cross-class comparison to fully leverage deep structural information during the KD phase?*

To address these challenges, inspired by curriculum learning [2, 51], we propose Mul**T**i-o**R**der Orchestrated C**U**rriculum Di**ST**illation (TRUST), a novel model-heterogeneous FGL framework. For Problem **I**), we introduce the **Progressive Curriculum Node Scheduler (PCNS)**, which progressively schedules node samples for each client from easy (nodes with typical class representations) to difficult (nodes situated near class boundaries that may confuse proxy models), thereby enabling heterogeneous

models to first absorb low-order semantic cues and then incrementally acquire high-order topological knowledge. For Issue **II)**, we propose the **Adaptive Curriculum Distillation Modulator (ACDM)**, a module that dynamically calibrates the distillation temperature during training—this mechanism allows the framework to fluidly shift emphasis between capturing global structures and preserving fine-grained node details. For Problem **III)**, in addition to the KL-Divergence loss, we introduce **Wasserstein-Driven Affinity Distillation (WDAD)**, which leverages class prototypes from private models and computes cross-class relational distances via the Wasserstein metric, thereby enabling comprehensive cross-class topological knowledge transfer. The contributions are as follows.

❶ *Problem Identification.* We are the first to systematically study **model-heterogeneous FGL** and formally characterize three core challenges in KD-based methods: the need for dynamic task difficulty, adaptive distillation signaling, and cross-class relational transfer.

❷ *Practical Solution.* We develop a curriculum-guided distillation framework that progressively schedules node difficulty, dynamically adjusts distillation strength, and enables knowledge transfer across classes, effectively reconciling heterogeneous architectures with complex graph topologies.

❸ *Experimental Validation.* We conduct extensive experiments on multiple graph benchmarks under diverse architecture heterogeneity settings. Empirical results demonstrate that `TRUST` consistently outperforms state-of-the-art baselines.

## 2   Preliminaries

### 2.1   Notations

**Graph Neural Networks.** Given a graph $\mathcal{G} = (\mathcal{V}, \mathcal{E})$, where $\mathcal{V}$ denotes the set of nodes and $\mathcal{E}$ represents the set of edges. For every node $v_i \in \mathcal{V}$, $v_i$ is associated with a $k$-dimensional feature vector $x_i$. The feature vectors of all nodes are represented collectively as the feature matrix $\mathbf{X} \in \mathbb{R}^{N \times k}$. The topology of the graph $\mathcal{G}$ is encoded in the adjacency matrix $\mathbf{A} \in \mathbb{R}^{N \times N}$ where $\mathbf{A}(v, u) = 1$ if $(v, u) \in \mathcal{E}$ and $\mathbf{A}(v, u) = 0$ otherwise. GNN models operate through iteratively updating node representations using a message-passing mechanism, where each node aggregates information from its local neighborhood and updates its own state. Let $\mathcal{N}_i$ denote the neighborhood nodes of $v_i$, and for the $l$-th layer of a GNN model, the representation of $h_i^l$ of node $v_i$ can be computed as:

$$h_i^l = \text{Update}\Big(h_i^{l-1}, \text{Aggregate}\big(\{h_j^{l-1} | v_j \in \mathcal{N}_i\}\big)\Big), \tag{1}$$

where $h_i^l$ denotes the representation of node $v_i$ at layer $l$.

**Knowledge Distillation** Traditional knowledge distillation methods employ the Kullback-Leibler (KL) Divergence loss to align the output distributions of student and teacher models:

$$L_{KL}(p^T, p^S) = \sum_i p^T(i) \log \frac{p^T(i)}{p^S(i)}, \tag{2}$$

where $p^T$ and $p^S$ denote the class probability distributions predicted by the teacher and student model respectively. These are computed via softmax function $\sigma$ and distillation temperature $\tau$:

$$p^T = \sigma(\frac{h^T}{\tau}), \quad p^S = \sigma(\frac{h^S}{\tau}). \tag{3}$$

**Model Heterogeneous Federated Graph Learning framework.** Let $S$ denotes the central server and $C^k$ denotes the $k$-th client with $K$ clients in total. Each client k holds its own graph $\mathcal{G}^k = (\mathcal{V}^k, \mathcal{E}^k)$. In a model heterogeneous setting, each client trains a model $w^k$ parameterized by $\theta^k$ on its own training data and then uploads it to the server, and the client models $\{w^1, w^2, ..., w^K\}$ do not share identical architectures. Mathematically, the learning objective can be formulated as:

$$\min_\theta \sum_{k=1}^K \frac{|\mathcal{V}^k|}{|\mathcal{V}|} L^k(y, w^k(\mathcal{G}^k)), \tag{4}$$

where $|\mathcal{V}^k|$ and $|\mathcal{V}|$ represents the number of samples of the client $k$ and the total number of samples across all clients respectively, $y$ denotes ground truth labels, and $\text{L}(\cdot, \cdot)$ is the empirical loss.

### 2.2   Model Heterogeneous FGL with proxy model

Inspired by FedType [49], we leverage proxy models to bridge heterogeneous models. In our framework, each client maintains both its private model and a proxy model. The proxy model is a small model with an identical architecture across all clients. The training process consists of three key phases: (1) Forward Distillation: First, each client distills knowledge from its private model to its

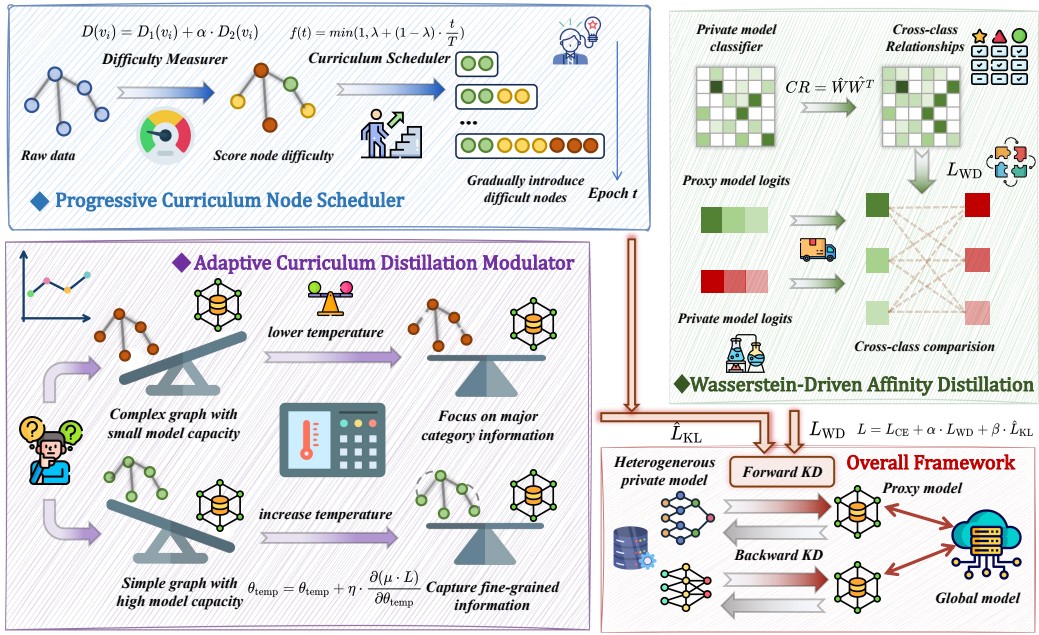

Figure 2: Architecture overview of `TRUST`, which includes three core components: (1) **PCNS** is a difficulty-progressive curriculum learning module. (2) **ACDM** provides capability-aware dynamic temperature adjustment. (3) **WDAD** employs Wasserstein-based cross-class distillation.

proxy model. (2) Global Aggregation: Then the weights of all proxy models are transmitted to the server where they are aggregated through weighted averaging to form a global model. (3) Backward Distillation: Next, the updated global model is distributed back to proxy models, which conducts knowledge distillation to transfer global knowledge to private models using a conformal model. The details of backward distillation are given in Section E in Appendix. This cyclic process continues until the global model converges. However, FedType is originally tailored for Euclidean data types as it does not fully exploit unique topological information of graph data. In order to generalize to graph data, our approach specifically addresses the three key challenges identified in Section **1**.

# 3 Methodology

## 3.1 Framework Overview

In this section, we present an overview of `TRUST`. `TRUST` adds three key components to the knowledge distillation process: (1) At the client side, during knowledge transfer from private to proxy models, we propose a PCNS to gradually introduce challenging nodes to proxy models. (2) At the same time, distillation temperature automatically calibrates throughout the process to adjust to both client model capabilities and local graph complexity. (3) After forward propagation, we introduce a WDAD loss, which incorporates cross-class comparison through optimal transport theory, preserving topological information that conventional distillation methods typically ignore. These components work synergistically to address the unique challenges of model-heterogeneous federated learning on graph data, while maintaining the privacy-preserving benefits of the federated paradigm. The framework is illustrated in Figure 2.

## 3.2 Progressive Curriculum Node Scheduler

**Motivation.** As established in Sec. 1, graph-structured data contains complex topological relationships. Therefore, premature exposure to this intricate high-order information during initial training phases results in client models struggling to learn generalizable patterns from highly complex nodes, and nodes situated near class boundaries with ambiguous class representations may even lead to misleading learning signals. Therefore, we need to design a progressive learning framework to introduce complex graph knowledge step by step.

**Curriculum Learning.** Drawing inspiration from human cognitive development, Curriculum Learning (CL) enables models to learn from data in a structured manner, transitioning from simpler to more complex samples, rather than processing all data uniformly in each epoch [2]. Prior works have proved that CL's effectiveness in improving model convergence, generalization, and final performance [23]. To implement this approach, we need to (1) make a formal definition of sample "difficulty" and (2) schedule data samples based on the proposed definition.

**Difficulty Measurer.** In GNN, each layer aggregates neighboring nodes to update node representations, as formalized in Equation 1. Intuitively, GNN models are more proficient at learning nodes whose neighbors belong to the same class because they share certain features. By contrast, for cross-class nodes connected to neighbors with divergent labels, GNN models receive conflicting gradient updates during aggregation, resulting in impaired learning. Therefore, we can quantify node learning difficulty through the neighborhood label distribution, where for a node $v_i$, the node difficulty is formally defined as the entropy of the distribution:

$$P_c(v_i) = \frac{|\{y_n = c \mid n \in \mathcal{N}_i \cup \{v_i\}\}|}{|\mathcal{N}_i \cup \{v_i\}|},$$
$$D_1(v_i) = -\sum_{c \in C} P_c(v_i) \log(P_c(v_i)), \tag{5}$$

where $P_c(v_i)$ computes proportion of class $c$ in neighborhood $\mathcal{N}_i \cup \{v_i\}$, $y_n$ denotes the label of node $n$, and $C$ represents the set of labels.

Notably, the neighbors of node $v_i$ may include samples from both the training and test datasets, and theoretically, the labels of test dataset neighbors are unavailable during the training phase, which means we need a locally pretrained GNN to generate pseudo-labels for unlabeled nodes. However, this approach introduces a fundamental challenge in decentralized FGL settings: locally pretrained GNN models may be unreliable due to the lack of global knowledge, potentially resulting in pseudo-label inaccuracies. Therefore, our difficulty measurer must exhibit robustness to potential label noise. Intuitively, we can implement this by evaluating the alignment between node representations and class prototypes. For a node $v_i$ with pseudo-label $y_i$, if its node representation $h_i$ demonstrates high similarity with prototype of class $y_i$, it is less likely to be assigned a wrong label because its features exhibit strong class-typical characteristics. But if it's the opposite, we can identify it as a "difficult" node because it has ambiguous node representation and the pretrained model has low confidence in the predicted pseudo-label. We can summarize this difficulty measurer as:

$$p_{y_i} = \frac{1}{|V_{y_i}|} \sum_{v \in V_{y_i}} h_v,$$
$$D_2(v_i) = 1 - \frac{\exp(h_i \cdot p_{y_i})}{\max_{c \in C} \exp(h_i \cdot p_c)}, \tag{6}$$

where $V_{y_i}$ is a subset of $V$ with all nodes belonging to class $y_i$ in it, and $p_{y_i}$ denotes the prototypes of class $y_i$ (mean embedding of nodes in $V_{y_i}$). Combining these two difficulty measurer, the overall node difficulty score can be formalized as follows:

$$D(v_i) = D_1(v_i) + \alpha \cdot D_2(v_i), \tag{7}$$

where $\alpha$ balances the weight of $D_2(v_i)$.

**Curriculum scheduler.** Once data samples are sorted by node difficulty in ascending order, each client implements a curriculum scheduler to gradually expose proxy model to more complex samples. The scheduler regulates the proportion of sorted training data used at each epoch $t$ through a pacing function. For simplicity, we adopt a linear function:

$$f(t) = \min(1, \lambda + (1 - \lambda) \cdot \frac{t}{T}), \tag{8}$$

where $\lambda$ denotes the available proportion of data samples at epoch 0, and $T$ is the epoch when full training data is first utilized. $f(t)$ monotonically increases from $\lambda$ to 1 over $T$ epochs. Notably, after $f(t)$ reaches 1, the model should still continue training for several additional epochs to ensure complete assimilation of challenging knowledge patterns.

Incorporating the proposed curriculum scheduler, we can reformulate the KL divergence loss as:

$$\hat{L}_{\text{KL}}(p^T, p^S) = \frac{1}{B} \sum_{j=1}^{B} \sum_i p_j^T(i) \log \frac{p_j^T(i)}{p_j^S(i)}, \tag{9}$$

where, for a given client $k$, the nodes are pre-sorted as $v_1, v_2, \ldots, v_{|\mathcal{V}^k|}$ in ascending order of their difficulty $D(v_i)$. At epoch $t$, the number of nodes selected from the beginning of this sorted list is $B_t = \lfloor f(t) \cdot |\mathcal{V}^k| \rfloor$. In Equation 9, $B$ corresponds to this $B_t$ at epoch $t$.

## 3.3 Adaptive Curriculum Distillation Modulator

**Motivation.** In model-heterogeneous FGL, each client adjusts local model architecture based on its available computational resources, and each client operates on a subgraph with varying complexity. Fixed temperature fails to account for differences in client model capabilities and topological knowledge complexity, limiting the ability of proxy models to learn intricate graph structures. To address this issue, we need to propose an ACDM that dynamically calibrates temperature.

**Adaptive Temperature Modulator.** To dynamically adjust distillation signals during training, intuitively if the current task difficulty is too low for a client model, increasing the difficulty can fully exploit model capability. Conversely, if the task exceeds the model's current capacity, reducing the difficulty prevents ineffective training, which resembles an adversarial process. Inspired by Generative Adversarial Networks (GANs) [25, 8], we implement this mechanism by converting the constant temperature value into a learnable parameter $\theta_{temp}$, which is optimized in the opposite direction of the client model parameters. In this way, $\theta_{temp}$ controls the difficulty of the loss minimization process, thereby enabling indirect dynamic adjustment of the distillation difficulty. Mathematically, the learning objective can be formulated as:

$$\theta_{\text{model}} = \theta_{\text{model}} - \eta \cdot \frac{\partial L}{\partial \theta_{\text{model}}}, \quad \theta_{\text{temp}} = \theta_{\text{temp}} + \eta \cdot \frac{\partial L}{\partial \theta_{\text{temp}}}, \tag{10}$$

where $\theta_{\text{model}}$ denotes the client model parameters (the proxy model more precisely in this framework), $\theta_{\text{temp}}$ is the temperature parameter, and $\eta$ denotes the learning rate.

**Curriculum Distillation Modulator.** While the Adaptive Temperature Modulator provides dynamic adjustment, we decide to prevent excessive interference at the early training stage since the model is still initializing and has limited learning capacity. Therefore, we incorporate curriculum learning to progressively increase the influence of the modulator. Specifically, we scale the loss $L$ by a factor $\mu$:

$$\theta_{\text{temp}} = \theta_{\text{temp}} + \eta \cdot \frac{\partial(\mu \cdot L)}{\partial \theta_{\text{temp}}}. \tag{11}$$

Here, $\mu$ is determined by the pacing function, which smoothly increases from 0 to 1 over the training epochs. We implement the pacing function as a cosine scheduler to progress more smoothly:

$$\mu = \frac{1 - cos(\frac{min(t,T)}{T} \cdot \pi)}{2}, \tag{12}$$

where $t$ denotes the current epoch.

**Module Pipeline.** Having established the optimization objective of ACDM, we now formalize its overall pipeline. The module can be conceptualized as a network layer denoted as $l_{temp}$ parameterized by $\theta_{temp}$. Each training epoch executes the following steps: (1) we first compute the scaling factor $\mu$ using Equation 12. (2) During forward propagation, $l_{temp}$ takes $\mu$ as input and outputs $\theta_{temp}$ to compute the distillation temperature $\tau$ for current epoch. (3) In backward propagation, $l_{temp}$ updates $\theta_{temp}$ via gradient descent using Equation 11. This cyclic process continues throughout the entire training phase. Notably, rather than directly use $\theta_{temp}$ as the distillation temperature $\tau$, we constrain it to a reasonable range via a sigmoid function. Therefore, KL-divergence loss is reformulated as:

$$\hat{L}_{\text{KL}}(p^T, p^S) = \frac{1}{B} \sum_{j=1}^{B} \sum_i \hat{p}_j^T(i) \log \frac{\hat{p}_j^T(i)}{\hat{p}_j^S(i)},$$

$$\hat{p}_j^T = \sigma(\frac{h_i^T}{\tau}), \quad \hat{p}_j^S = \sigma(\frac{h_i^S}{\tau}), \tag{13}$$

$$\tau = \tau_{min} + \tau_{max} \cdot sigmoid(\theta_{temp}),$$

where $\tau_{min}$ and $\tau_{max}$ are the upper bound and lower bound for distillation temperature.

## 3.4 Wasserstein-Driven Affinity Distillation

**Motivation.** As shown in Equation 2, KL-divergence only compares intra-class probability distributions. For graph data with complex topological relationships, this approach may fail to capture

Table 1: **Comparison with the state-of-the-art methods** on five real-world datasets under moderate heterogeneity. For each dataset, we report local and global accuracy(%) (with red/green markers indicating regression/improvement over FedAvg). The best and second-best results are marked with **bold** and underline, respectively. Additional results under more settings are in Appendix D.

| Category | Methods | Cora | | CiteSeer | | PubMed | | CS | | Photo | |
|---|---|---|---|---|---|---|---|---|---|---|---|
| | acc Type | local | global | local | global | local | global | local | global | local | global |
| FL | FedAvg [ASTAT17] | 81.36 | 64.52 | 82.61 | 65.48 | 88.10 | 82.09 | 90.10 | 83.35 | 90.14 | 84.10 |
| | FedNOVA [NeurIPS20] | $81.54_{\uparrow 0.18}$ | $64.97_{\uparrow 0.45}$ | $82.76_{\uparrow 0.15}$ | $66.22_{\uparrow 0.74}$ | $88.20_{\uparrow 0.10}$ | $82.87_{\uparrow 0.78}$ | $90.13_{\uparrow 0.03}$ | $82.37_{\downarrow 0.98}$ | $90.34_{\uparrow 0.20}$ | $86.70_{\uparrow 2.60}$ |
| | FedProto [AAAI22] | $79.17_{\downarrow 2.19}$ | $64.79_{\uparrow 0.27}$ | $82.61_{\uparrow 0.00}$ | $67.24_{\uparrow 1.76}$ | $88.10_{\uparrow 0.00}$ | $83.46_{\uparrow 1.37}$ | $91.97_{\uparrow 1.87}$ | $80.81_{\downarrow 2.54}$ | $86.38_{\downarrow 3.76}$ | $84.04_{\downarrow 0.06}$ |
| | MOON [CVPR21] | $81.52_{\uparrow 0.16}$ | $65.70_{\uparrow 1.18}$ | $81.58_{\downarrow 1.03}$ | $63.84_{\downarrow 1.64}$ | $88.15_{\uparrow 0.05}$ | $82.34_{\uparrow 0.25}$ | $91.78_{\uparrow 1.68}$ | $83.81_{\uparrow 0.46}$ | $90.40_{\uparrow 0.26}$ | $86.18_{\uparrow 2.08}$ |
| | FedType [ICML24] | $82.25_{\uparrow 0.89}$ | $72.96_{\uparrow 8.44}$ | $83.20_{\uparrow 0.59}$ | $64.24_{\downarrow 1.24}$ | $87.39_{\downarrow 0.71}$ | $84.34_{\uparrow 2.25}$ | $91.64_{\uparrow 1.54}$ | $86.36_{\uparrow 3.01}$ | $88.91_{\downarrow 1.23}$ | $90.17_{\uparrow 6.07}$ |
| FGL | AdaFGL [ICDE24] | $81.93_{\uparrow 0.57}$ | $64.61_{\uparrow 0.09}$ | $80.40_{\downarrow 2.21}$ | $65.48_{\uparrow 0.00}$ | $85.35_{\downarrow 2.75}$ | $82.87_{\uparrow 0.78}$ | $91.11_{\uparrow 1.01}$ | $84.24_{\uparrow 0.89}$ | $90.64_{\uparrow 0.50}$ | $82.54_{\downarrow 1.56}$ |
| | FedGTA [VLDB24] | $77.73_{\downarrow 3.63}$ | $64.43_{\downarrow 0.09}$ | $79.96_{\downarrow 2.65}$ | $66.18_{\uparrow 0.70}$ | $85.62_{\downarrow 2.48}$ | $82.57_{\uparrow 0.48}$ | $91.80_{\uparrow 1.70}$ | $83.89_{\uparrow 0.54}$ | $90.87_{\uparrow 0.73}$ | $84.56_{\uparrow 0.46}$ |
| Model-Heterogeneous FGL | TRUST | $\mathbf{83.90_{\uparrow 2.54}}$ | $\mathbf{75.32_{\uparrow 10.80}}$ | $\mathbf{85.84_{\uparrow 3.23}}$ | $\mathbf{67.43_{\uparrow 1.95}}$ | $\mathbf{89.06_{\uparrow 0.96}}$ | $\mathbf{84.57_{\uparrow 2.48}}$ | $\mathbf{92.19_{\uparrow 2.09}}$ | $\mathbf{87.07_{\uparrow 3.72}}$ | $\mathbf{91.42_{\uparrow 1.28}}$ | $\mathbf{91.50_{\uparrow 7.40}}$ |

subtle but important cross-class structural dependencies. To address this limitation, we propose a cross-class comparison mechanism.

**Cross-class Relationships.** We quantify cross-class relationships (CR) by computing cosine similarity between prototypes of different categories. Following Equation 6, prototypes can be derived from the mean node embeddings for each category. Drawing inspiration from [37], for implementation efficiency we decide to instead adopt the classifier weight vectors of the private model to compute class prototypes. For categories with certain connections, their classifier weight vectors also share certain similarity. Let $W \in \mathbb{R}^{c \times n}$ denotes the classifier weight matrix, where $c$ is the number of classes and $n$ is the feature dimensions. The CR metric is computed as:

$$\hat{w}_i = \frac{w_i}{||w_i||_2}, \quad CR = \hat{W}\hat{W}^T, \tag{14}$$

where $w_i$ represents the weight vector for class $i$ in $W$, $\hat{w}_i$ denotes the $L_2$-normalized weight vector of $w_i$, and CR captures the pairwise similarity between all classes.

**Wasserstein-Driven Affinity Distillation Loss.** To quantify the probability distribution difference between $p^T$ and $p^S$, based on Wasserstein distance [4], we define WDAD loss as:

$$L_{\text{WD}}(p^T, p^S) = \min \sum_{i,j} c_{ij} q_{ij} + \eta \cdot q_{ij} \log q_{ij}, \tag{15}$$

where $q_{ij}$ and $c_{ij}$ represents the transmitted probability mass and transmission cost from teacher (private model in forward distillation) category i to student (proxy model) category j respectively, and $\eta$ is a regularization hyper-parameter. $q_{ij}$ is constrained by:

$$\sum_j q_{ij} = p_i^T, \quad \sum_i q_{ij} = p_j^S, \quad q_{ij} \geq 0. \tag{16}$$

Intuitively, if two categories are similar in the feature space, the transmission cost $c_{ij}$ should be lower. Therefore, we can leverage CR to compute $c_{ij}$ using a Gaussian kernel:

$$c_{ij} = 1 - \exp(-\kappa(1 - CR_{ij})), \tag{17}$$

where $\kappa$ is a hyper-parameter adjusting the sensitivity to CR. Overall, WDAD loss explicitly introduces cross-class relationships through transmission cost $c_{ij}$. By minimizing this loss, we enforce consistency between private and proxy models in their probability allocation, particularly for categories that are similar in the feature space.

Moreover, rather than completely replace $\hat{L}_{\text{KL}}$ with $L_{\text{WD}}$, we decide to employ a weighted combination of $L_{\text{WD}}$ and $\hat{L}_{\text{KL}}$ to ensure smooth transition between the two objectives. Therefore, our **final optimization objective** can be formulated as:

$$L = L_{\text{CE}} + \alpha \cdot L_{\text{WD}} + \beta \cdot \hat{L}_{\text{KL}}, \tag{18}$$

where $L_{\text{CE}}$ denotes the cross-entropy function for classification tasks, and $\alpha$ and $\beta$ are two hyper-parameters regulating the weight of $L_{\text{WD}}$ and $\hat{L}_{\text{KL}}$ respectively.

## 4 Experiment

In this section, we comprehensively evaluate TRUST through four axes: **Q1** (Superiority), **Q2** (Resilience). **Q3** (Effectiveness), **Q4** (Sensitivity),

## 4.1 Experimental Setup

**Datasets.** To effectively evaluate the performance of our approach, we employed five benchmark graph datasets of various scales and distributions, including Cora [31], CiteSeer [7], PubMed [38], CS, and Photo. Detailed descriptions and splits for these datasets can be found in Appendix C.1. Moreover, the implementation details and parameter settings can be found in Appendix C.3.

**Counterparts.** We compare TRUST against several traditional FL methods: (1) **FedAvg** [ASTAT17] [32], (2) **FedNOVA** [NeurIPS20] [47], (3) **FedProto**[AAAI22] [39], (4) **MOON** [CVPR 21][26],(5) **FedType** [ICML24] [49]; two popular FGL approaches: (6) **AdaFGL** [ICDE24] [28]; (7) **FedGTA** [VLDB24] [29].Detailed descriptions can be found in Appendix C.2.

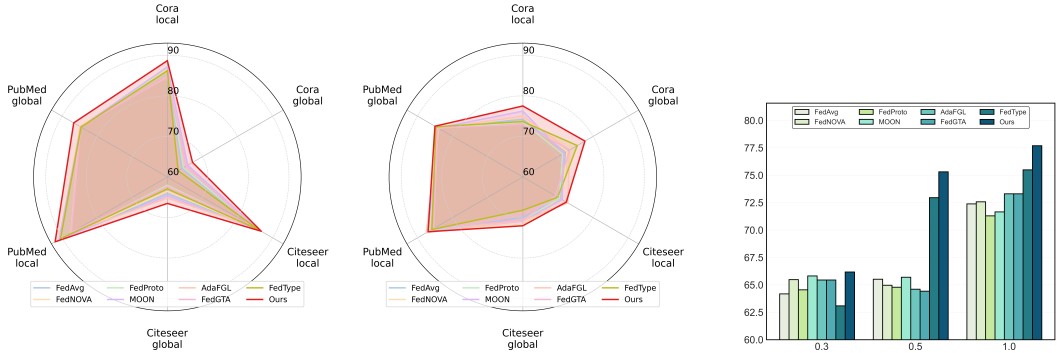

Figure 3: We report the performance of different methods under varying data heterogeneity levels on Cora, CiteSeer, and PubMed, with both local and global accuracy. The red color denotes the performance of TRUST. (*First*): severe heterogeneity($\alpha$=0.3). (*Second*): mild heterogeneity($\alpha$=1.0). (*Third*): Results on Cora under different heterogeneity levels with $\alpha$ set to 0.3, 0.5, and 1.0.

## 4.2 Superiority

To answer **Q1**, we conducted systematic experimental evaluations in a variety of subgraph data heterogeneity environments, which we control by partitioning the graph using a Dirichlet distribution with parameter $\alpha$. Smaller values of $\alpha$ lead to more extreme subgraph distributions. We compared the performance of TRUST with existing approaches based on two metrics: local accuracy and global accuracy. Local accuracy is measured as the average classification accuracy across all clients, reflecting the model's effectiveness on decentralized subsets. Global accuracy is evaluated on the aggregated model at the server. For most existing methods, the global model is updated via knowledge distillation. In contrast, both TRUST and FedType update the global model by simply averaging the parameters of the proxy models. The results are summarized in Tab. 1 and Figure 3.

From the table, several key observations can be made (**Obs.**): **Obs. ❶Existing approaches show suboptimal performance in heterogeneous FGL scenarios.** For instance, when $\alpha$ is set to 0.3, most existing methods achieve accuracy on the Cora dataset that is either lower than or comparable to FedAvg. Notably, when $\alpha$ is reduced to 0.1, the performance of these methods drops significantly, with the average global accuracy consistently falling below 35%.

**Obs. ❷ TRUST demonstrates remarkable robustness across various graph data heterogeneity scales.** Under moderate heterogeneity conditions ($\alpha = 0.5$), TRUST exhibits clear advantages. As shown in Tab. 1, TRUST consistently outperforms both FL and FGL baseline methods across different datasets in terms of both local and global accuracy. For example, TRUST achieves a global accuracy of 75.32% on the Cora dataset, surpassing the best baseline method, FedType (72.96%) by 2.36%, and outperforming FedAvg (64.52%) by a significant 10.80% margin. Furthermore, as shown in Figure 3, TRUST consistently outperforms all baselines across various heterogeneity levels. In highly heterogeneous environments,TRUST achieves varying degrees of performance improvement over all baselines .In mildly heterogeneous settings, it demonstrates an average accuracy gain of 2.25%.

## 4.3 Resilience

To address **Q2**, we evaluate the performance of each method on the Cora dataset across varying levels of data heterogeneity, where $\alpha$ is set to 0.3, 0.5, and 1.0.Figure 3(Third) illustrates that TRUST

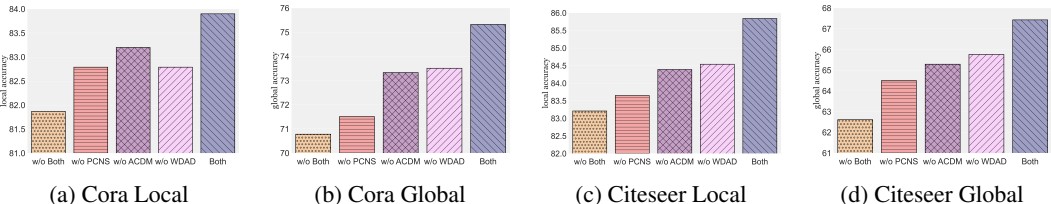

|            (a) Cora Local            |            (b) Cora Global            |            (c) Citeseer Local            |            (d) Citeseer Global            |

Figure 4: **Ablation Study** of the key components PCNS,ACDM and WDAD on Cora and Citeseer datasets. For an in-depth analysis, please refer to Sec. 4.4.

achieves robust performance gains across varying levels of data heterogeneity, outperforming other algorithms by an average of 5.78%, and at $\alpha = 0.5$ on the Cora dataset, it even surpasses FedAvg by 10.80%.This demonstrates that `TRUST` can effectively identify heterogeneous graphs and maintain superior performance, even under challenging conditions with extreme data heterogeneity.

## 4.4  Effectiveness

To address **Q3**, we conduct an ablation study on the key components of the framework: PCNS, ACDM, and WDAD. Experimental results on Cora and CiteSeer are presented in Figure 4.

The ablation study demonstrates that all three components contribute substantially to performance improvement. PCNS has the most pronounced individual effect; its removal results in a 2.9% drop in accuracy (Citeseer Global: 67.43% $\rightarrow$ 64.51%), confirming its effectiveness in progressive node scheduling. ACDM provides dataset-dependent benefits, yielding a 1.93% gain on Citeseer Global through dynamic temperature modulation. WDAD consistently contributes 0.8–1.6% improvements by enabling cross-class knowledge transfer.

When PCNS, ACDM, and WDAD are combined, the model achieves the best performance, effectively distilling both structural and semantic knowledge into a well-generalizable student model.

## 4.5  Sensitivity

To address **Q4**, we conduct analyses on hyperparameters of `TRUST`. Specifically, we analyze the model's performance under different values of $\lambda$ and $T$, as defined in Equation 8.We evaluate all combinations of $\lambda \in \{0.25, 0.5, 0.75\}$ and $T \in \{20, 40, 80, 100\}$. Results shown in Figure 5 demonstrate that the choice of hyperparameters $\lambda$ and $T$ has a minimal impact on the performance of `TRUST`, proving the robustness of `TRUST`.

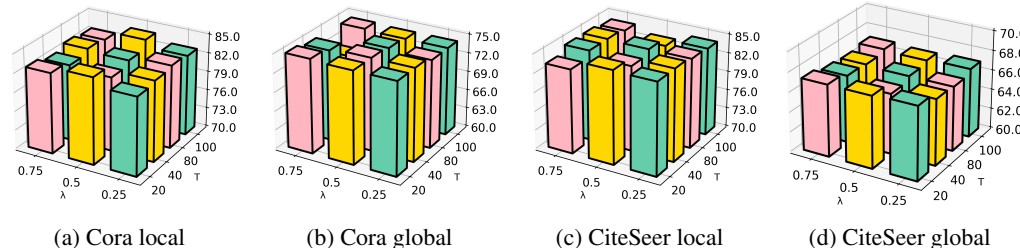

|            (a) Cora local            |            (b) Cora global            |            (c) CiteSeer local            |            (d) CiteSeer global            |

Figure 5: **Analysis on hyper-parameter** in `TRUST`.Node classification results are evaluated on Cora and CiteSeer datasets under various hyperparameter combinations, testing both global and local accuracy. All experiments are conducted under the setting of $\alpha = 0.5$.

## 5  Conclusion

In this paper, we propose `TRUST` to address model heterogeneity in Federated Graph Learning. Based on knowledge distillation to bridge heterogenous client models, we integrates three key strategies to effectively handle complex topological information in graph-structured data. We first employ PCNS to progressively introduce complex samples based on learning difficulty. Then we propose ACDM for dynamic temperature adjustment. We further propose WDAD that captures cross-class structural relationships. Comprehensive experiments across five benchmark datasets demonstrate `TRUST`'s

superior capability in resolving model heterogeneity challenges while preserving graph topological properties. The framework establishes a state-of-the-art for heterogeneous FGL systems.

## Acknowledgement

This work is supported by National Natural Science Foundation of China under Grant (62361166629, 62225113, 623B2080), the Major Project of Science and Technology Innovation of Hubei Province (2024BCA003, 2025BEA002), and the Innovative Research Group Project of Hubei Province under Grants 2024AFA017. The supercomputing system at the Supercomputing Center of Wuhan University supported the numerical calculations in this paper.

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

## A Notations

We present a comprehensive review of the commonly used notations and their definitions in Tab. 2.

Table 2: Notation and Definitions

| Notation | Definition |
|---|---|
| $K$ | The number of clients. |
| $\mathcal{D}^k$ | The dataset of k-th client. |
| $w^k$ | The model of k-th client. |
| $\mathcal{G}$ | Graph data. |
| $\mathcal{V}$ | The node set of $\mathcal{G}$. |
| $\mathcal{E}$ | The edge set of $\mathcal{G}$. |
| $C$ | The label set of $\mathcal{G}$. |
| $\mathbf{X}$ | The feature matrix of $\mathcal{G}$. |
| $\mathbf{A}$ | The adjacency matrix of $\mathcal{G}$. |
| $v_i$ | Node $i$ in $\mathcal{V}$. |
| $x_i$ | Feature vector of node $i$. |
| $y_i$ | Label of node $i$. |
| $\mathcal{N}_i$ | The neighborhood nodes of $v_i$. |
| $h_i^l$ | The representation of $v_i$ at the $l$-th layer of GNN. |
| $D(v_i)$ | The node difficulty of $v_i$. |
| $V_c$ | The node subset consisting of all nodes labeled as c. |
| $p_c$ | The prototype of class c. |
| $\lambda$ | The available proportion of data samples at epoch 0. |
| $T$ | The epoch when full training data is first utilized. |
| $\eta$ | The learning rate. |
| $\mu$ | The scaling factor of Adaptive Temperature Modulator. |
| $p^T$ | The class probability distribution predicted by the teacher model. |
| $p^S$ | The class probability distribution predicted by the student model. |
| $\tau$ | The distillation temperature. |
| $CR$ | The pairwise similarity matrix between all classes. |
| $c_{ij}$ | The probability mass transferred from teacher category i to student category j. |
| $q_{ij}$ | The transmission cost from teacher category i to student category j. |
| $\kappa$ | The hyper-parameter adjusting the sensitivity to cross-class similarity in Gaussian kernel. |
| $L_{\text{CE}}$ | The cross-entropy function for classification tasks. |
| $L_{\text{WD}}$ | The proposed Wasserstein-Driven Affinity Distillation loss. |
| $L_{\text{KL}}$ | The Kullback-Leibler (KL) Divergence loss. |

## B Related Work.

**Federated Graph Learning.** With recent advances in Federated Learning (FL) for vision and language tasks [20, 18], Federated Graph Learning (FGL) further extends FL to graph-structured data [41, 42]. Existing FGL-related researches are mainly focused on processing graph-structured data. Based on how graphs are distributed across clients, these methods can fall into three categories: graph-level, subgraph-level and node-level [12]. Graph-level FGL methods consider settings where clients possess completely disjoint graphs while in subgraph-level FGL settings, each client holds a subgraph that is part of a larger global graph [57, 52]. In node-level FGL, each agent possesses the ego-networks of one or multiple nodes [35, 12]. However, these methods often assume homogeneous model architectures cross clients [17, 54, 40], which is an impractical constraint that degrades performance in real-world scenarios. To tackle this problem, we propose a model-heterogeneous method that transfers knowledge between clients in a model-agnostic manner through a novel knowledge distillation framework, overcoming limitations of existing FGL approaches.

**Model-heterogeneous Federated Learning.** Model-heterogeneous federated learning entails learning from others without disclosing information about local model architectures. Recent works on model-heterogeneous federated learning can be categorized into three approaches: data-level,

model-level and server-level [54]. Data-level methods, such as TAKFL [34] and DESA [15], distill knowledge via external public data distributions. Model-level methods, such as pFedHR [48] and pFedClub [46], share partial model structures or reassembled components to other clients. Server-level methods, such as FedMRL [55] and FedType [49], deploy proxy models as intermediaries to bridge heterogeneous models. While effective for Euclidean data, these methods fail to preserve graph topological features during knowledge transfer. Our work breaks this limitation by uniquely integrating curriculum learning and Wasserstein distance to capture complex structural information, improving graph representation in model-heterogeneous FGL.

**Knowledge Distillation.** Knowledge Distillation (KD) is a machine learning method designed for model compression and knowledge transfer [13, 14]. Traditional KD transfers knowledge from a large teacher model to a compact student model through softened outputs or intermediate representations [13]. Recently, researches have demonstrated its effectiveness in facilitating model collaboration in federated learning (FL), particularly in scenarios involving heterogeneous models or data distributions [54]. For instance, TAKFL [34] introduces KD to FL framework to distill knowledge from heterogeneous clients and then integrate the separately distilled knowledge with task arithmetic. Similarly, FedMKD [24] combines KD and attention mechanisms to work with model heterogeneity in FL. However, existing KD methods typically rely on KL-Divergence minimization, which only performs intra-category comparisons between teacher and student models. By contrast, our work proposes a noval Wasserstein knowledge distillation framework which introduces Wasserstein-distance to enable cross-class comparison, explicitly modeling graph-structured relationships during knowledge transfer and maintaining topological properties of the original graph data.

## C  Experimental Details.

### C.1  Dataset Details.

To assess the effectiveness of TRUST, we conduct experiments on five real-world graph datasets: Cora, CiteSeer, PubMed, Amazon-Photo and CoAuthor-CS. Each dataset is split into training, validation, and test sets in a fixed 20%/40%/40% ratio. The key statistics of these datasets are summarized in Tab. 3. A detailed description is provided below:

- **Cora, CiteSeer, and PubMed.** These three citation network datasets are standard benchmarks in graph-based machine learning, especially for tasks like node classification and link prediction. In these datasets, nodes correspond to academic papers, while edges represent citation links. Each node is assigned a class label, and its feature vector is constructed from textual information such as words in the title or abstract. These datasets exhibit sparsity and high dimensionality, making them well-suited for evaluating the effectiveness and scalability of graph neural networks (GNNs).
- **CoAuthor-CS.** This dataset represents a co-authorship network in the field of computer science, where nodes correspond to research papers, and edges denote co-authorship relations. Each paper is associated with a topic category, and features are extracted from the paper's title and abstract. This dataset is commonly used to evaluate node classification and community detection algorithms.
- **Amazon-Photo.** This dataset is built from the Amazon product catalog, where nodes represent product images and edges indicate co-purchase relationships. Each photo is categorized into a specific class, and node features are derived from image metadata. Amazon-Photo serves as a benchmark for testing graph-based learning models in visual domains.

Table 3: **Statistics** of datasets used in experiments.

| Dataset | #Nodes | #Edges | #Classes | #Features |
|---|---|---|---|---|
| Cora | 2,708 | 5,278 | 7 | 1,433 |
| Citeseer | 3,327 | 4,552 | 6 | 3,703 |
| Pubmed | 19,717 | 44,324 | 3 | 500 |
| Coauthor-CS | 18,333 | 327,576 | 15 | 6,805 |
| Amz-Photo | 7,650 | 287,326 | 8 | 745 |

### C.2  Counterpart Details.

This section provides a comprehensive overview of the baseline approaches employed in our study.

- **FedAvg** [ASTAT17]. A foundational algorithm in Federated Learning, FedAvg operates by allowing clients to independently train models on their local datasets and subsequently transmit their model updates to a central server. The server performs a weighted aggregation of these updates to refine the global model, which is then redistributed to the clients for further local training. By transmitting only model parameters instead of raw data, FedAvg reduces communication costs and enhances privacy. However, it struggles with performance degradation in scenarios where client data distributions are highly non-IID [27, 33].
- **FedNova** [NeurIPS20]. FedNova refines the FedAvg framework by introducing normalization to local updates before aggregation. Unlike standard averaging methods, FedNova ensures that each client's contribution to the global model is proportional to the amount of data it possesses. This approach addresses the issue of unequal client influence, leading to more balanced and efficient convergence. FedNova is particularly beneficial in federated environments where data distributions are skewed across clients.
- **FedProto** [AAAI22].FedProto introduces prototype-based federated learning to address both data and model heterogeneity across clients. Unlike gradient-based approaches, the framework exchanges class prototypes (mean feature representations) between server and clients, enabling knowledge transfer while accommodating different model architectures and non-IID data distributions. Through prototype aggregation and local regularization, FedProto achieves superior communication efficiency and convergence guarantees while preserving privacy [39]. The method demonstrates strong performance on image datasets while requiring significantly fewer communicated parameters than traditional FL approaches.
- **Moon** [CVPR21].MOON adopts a model-contrastive approach to address data heterogeneity in federated learning. The framework utilizes similarities between model representations to correct local training through model-level contrastive learning, providing an effective solution for collaborative training with deep learning models on image datasets while preserving data privacy.
- **FedType** [ICML24]. FedType[49] introduces a novel uncertainty-based asymmetrical reciprocity learning framework to address model heterogeneity in federated learning. The approach employs small identical proxy models as secure intermediaries for information exchange, eliminating the need for public data while ensuring privacy protection. Through bidirectional knowledge distillation with dynamic conformal prediction, FedType achieves superior performance across diverse model architectures and datasets, demonstrating significant improvements in communication efficiency and model security compared to existing methods.
- **AdaFGL** [ICDE24].AdaFGL introduces a novel paradigm for federated node classification with topology heterogeneity, addressing the critical challenge of structural divergence among clients in federated graph learning. The framework employs a decoupled two-step approach: first obtaining a federated knowledge extractor through collaborative training, then performing personalized propagation optimized by local topology. By incorporating adaptive mechanisms that automatically balance homophilous and heterophilous propagation based on quantified structural characteristics, AdaFGL achieves state-of-the-art performance across 12 benchmark datasets while minimizing communication overhead and privacy risks [28].
- **FedGTA** [VLDB24]. FedGTA is tailored for large-scale graph federated learning, tackling issues of slow convergence and suboptimal scalability. Unlike prior methods that focus on either optimization strategies or complex local models, FedGTA integrates topology-aware local smoothing with mixed neighbor feature aggregation to improve learning efficiency [29]. By leveraging graph structures in aggregation, it enhances scalability and performance in federated graph learning.

## C.3 Implementation Details.

The experiments are conducted on NVIDIA GeForce RTX 3090 GPUs, paired with dual Intel(R) Xeon(R) Gold 6240 CPUs @ 2.60GHz (36 cores per socket, Turbo Boost up to 3.90GHz). The deep learning framework used is PyTorch (v2.5.1) with CUDA 12.1.

The experimental setup involves 10 clients. To simulate real-world model heterogeneity, each client maintains a private model whose architecture is randomly selected from GCN, GAT, or GraphSAGE. All private models are configured with three layers, a hidden dimension of 64, and a dropout rate of 0.3. To facilitate collaboration, each client is equipped with an additional small proxy model that serves as a communication bridge. This proxy model employs a standardized GCN architecture with 3 layers to ensure compatibility across clients. On the server side, we implement a global model

Table 4: **Comparison with the state-of-the-art methods** on three selected real-world datasets. The alpha is set to 0.3 and 1.0. The best and second-best results are highlighted with **bold** and underline, respectively.

| Category | Methods | 0.3 alpha | | | | | | 1.0 alpha | | | | | |
|---|---|---|---|---|---|---|---|---|---|---|---|---|---|
| | | Cora | | CiteSeer | | PubMed | | Cora | | CiteSeer | | PubMed | |
| | | local | global | local | global | local | global | local | global | local | global | local | global |
| FL | FedAvg [ASTAT17] | 86.39 | 64.18 | 84.43 | 64.14 | 90.82 | 84.64 | 74.21 | 72.39 | 70.57 | 69.99 | 86.82 | 84.16 |
| | FedNOVA [NeurIPS20] | $86.58_{\downarrow0.19}$ | $65.48_{\uparrow1.30}$ | $85.18_{\uparrow0.75}$ | $63.54_{\downarrow0.60}$ | $90.92_{\downarrow0.10}$ | $84.99_{\uparrow0.35}$ | $74.77_{\downarrow0.56}$ | $72.58_{\uparrow0.19}$ | $71.17_{\uparrow0.60}$ | $\underline{71.77}_{\uparrow1.78}$ | $86.84_{\uparrow0.02}$ | $83.95_{\downarrow0.21}$ |
| | FedProto [IJCAI23] | $86.93_{\downarrow0.54}$ | $64.55_{\uparrow0.37}$ | $84.88_{\uparrow0.45}$ | $62.05_{\downarrow2.09}$ | $90.75_{\downarrow0.07}$ | $84.59_{\downarrow0.05}$ | $73.50_{\downarrow0.71}$ | $71.30_{\downarrow1.09}$ | $70.42_{\downarrow0.15}$ | $70.73_{\uparrow0.74}$ | $86.64_{\downarrow0.18}$ | $84.08_{\downarrow0.08}$ |
| | MOON [CVPR21] | $\underline{87.12}_{\downarrow0.73}$ | $\underline{65.82}_{\uparrow1.64}$ | $\underline{85.73}_{\uparrow1.30}$ | $64.69_{\uparrow0.55}$ | $90.85_{\uparrow0.03}$ | $84.69_{\uparrow0.05}$ | $\underline{76.04}_{\uparrow1.83}$ | $71.66_{\downarrow0.73}$ | $71.21_{\uparrow0.64}$ | $70.88_{\uparrow0.89}$ | $86.97_{\uparrow0.15}$ | $84.16_{\uparrow0.00}$ |
| | FedType [ICML24] | $86.21_{\downarrow0.18}$ | $63.09_{\uparrow1.09}$ | $85.72_{\uparrow1.29}$ | $62.95_{\uparrow1.19}$ | $90.44_{\downarrow0.38}$ | $84.59_{\downarrow0.05}$ | $73.66_{\downarrow0.55}$ | $\underline{75.50}_{\uparrow3.11}$ | $69.97_{\downarrow0.60}$ | $68.20_{\downarrow1.79}$ | $85.91_{\uparrow0.15}$ | $84.74_{\uparrow0.58}$ |
| FGL | AdaFGL [ICDE24] | $84.60_{\downarrow1.79}$ | $65.45_{\uparrow1.27}$ | $84.35_{\downarrow0.08}$ | $\underline{65.22}_{\uparrow1.08}$ | $87.52_{\downarrow3.30}$ | $84.89_{\uparrow0.25}$ | $74.96_{\uparrow0.75}$ | $73.31_{\uparrow0.92}$ | $\mathbf{72.82}_{\uparrow2.25}$ | $70.73_{\downarrow0.74}$ | $\mathbf{87.47}_{\uparrow0.65}$ | $84.03_{\downarrow0.13}$ |
| | FedGTA [VLDB24] | $84.72_{\downarrow1.67}$ | $65.45_{\uparrow1.27}$ | $83.66_{\downarrow0.77}$ | $62.35_{\downarrow1.79}$ | $87.30_{\downarrow3.52}$ | $84.67_{\uparrow0.03}$ | $72.94_{\downarrow1.27}$ | $73.31_{\uparrow0.92}$ | $69.26_{\downarrow1.31}$ | $70.58_{\uparrow0.59}$ | $85.39_{\uparrow1.43}$ | $84.16_{\uparrow0.00}$ |
| Model-Heterogeneous FGL | TRUST | $\mathbf{88.67}_{\uparrow2.28}$ | $\mathbf{67.18}_{\uparrow3.00}$ | $\mathbf{86.76}_{\uparrow2.33}$ | $\mathbf{66.52}_{\uparrow2.38}$ | $\mathbf{91.97}_{\uparrow1.15}$ | $\mathbf{86.67}_{\uparrow2.03}$ | $\mathbf{77.50}_{\uparrow3.29}$ | $\mathbf{77.70}_{\uparrow5.31}$ | $\underline{72.38}_{\uparrow1.81}$ | $\mathbf{72.03}_{\uparrow2.04}$ | $\underline{86.99}_{\uparrow0.17}$ | $\mathbf{85.04}_{\uparrow0.88}$ |

that also adopts a GCN backbone and uses a hidden dimension of 32 while sharing the remaining configurations with the client models.

To simulate data heterogeneity, client data is partitioned using a Dirichlet distribution [22], drawing $p_k \sim \mathrm{Dir}(\alpha)$ to allocate a fraction $p_k^c$ of class $c$ to client $k$. Each client's subgraph is split into training, validation, and test sets with a ratio of 0.6/0.2/0.2, respectively. We use the Adam optimizer with a learning rate of $5 \times 10^{-3}$ and a weight decay of $4 \times 10^{-4}$ for training. The number of communication rounds is set to 200.

For PCNS, in the Difficulty Measurer, we set $\alpha = 0.5$. In the Curriculum Scheduler, the hyperparameters $\lambda$ and $T$ are selected via grid search over $\{0.25, 0.5, 0.75\}$ and $\{20, 40, 80, 100\}$, respectively. For ACDM, the model parameters $\theta_{\mathrm{model}}$ are optimized using Adam (learning rate: 0.01, weight decay: $5 \times 10^{-4}$), while the temperature parameters $\theta_{\mathrm{temp}}$ are optimized using SGD (momentum: 0.9, weight decay: $4 \times 10^{-4}$). The conformal mode parameters in backward distillation follow the configuration in FedType. In WDAD, we set $\eta = 0.05$, $\kappa = 1.0$, with loss weights $\alpha = 0.025$ (Wasserstein) and $\beta = 0.01$ (KL divergence).

Regarding baseline implementations, all selected FL baselines, except for FedAvg, support model heterogeneity and operate similarly to their applications in computer vision tasks. However, for FGL methods, to the best of our knowledge, our work is the first to explore model heterogeneity in FGL settings. Consequently, the chosen FGL baselines are originally designed for model homogeneity only. To include them (along with FedAvg) in heterogenous settings, we remove direct parameter sharing between architecturally different models and maintain all other original components and hyperparameters. For evaluation, we distill a global model from local private models and assess its accuracy.

# D Additional Experimental Results.

## D.1 Comparison with More FGL Baselines.

To further validate the efficacy of our approach, we compare TRUST against two additional FGL baselines, FedTAD and FedSSL, on the Cora, Citeseer, Pubmed, and CS datasets. The corresponding results are presented in Tab. 5.

Table 5: Local and gobal accuracy results of TRUST and two additional FGL baselines on the Cora, Citeseer, Pubmed and CS datasets under moderate heterogeneity($\alpha = 0.5$).

| Methods | Cora | | Citeseer | | Pubmed | | CS | |
|---|---|---|---|---|---|---|---|---|
| | local | global | local | global | local | global | local | global |
| FedTAD | 80.27 | 65.70 | 83.24 | 66.37 | 87.88 | 83.30 | 90.72 | 82.93 |
| FedSSL | 79.53 | 64.79 | 82.32 | **68.15** | 88.07 | 83.58 | 91.97 | 80.73 |
| TRUST | **83.90** | **75.32** | **85.8**4 | 67.43 | **89.06** | **84.57** | **92.19** | **87.07** |

As shown in Tab. 5, TRUST achieves the best performance on 7 out of the 8 evaluation metrics. The only exception is the global accuracy on the Citeseer dataset, where FedSSL attains the best result. These comprehensive results further confirm the effectiveness of TRUST against state-of-the-art FGL methods.

## D.2 Comparison under Extreme Data Heterogeneity.

We provide additional local and global accuracy results under the $\alpha$ settings of 0.3 and 1.0 in Tab. 4, which is previously visualized in Figure 3. We also experiment at more extreme level of data heterogeneity, where $\alpha = 0.1$. Results are presented in Tab. 6.

Table 6: Comparison with state-of-the-art methods on three datasets under $\alpha = 0.1$. The best and second-best results are highlighted with bold and underline, respectively.

| Methods | Cora | | Citeseer | | Pubmed | |
|---|---|---|---|---|---|---|
| | local | global | local | global | local | global |
| FedAvg | 93.14 | 35.04 | 86.33 | 54.98 | 97.61 | 79.99 |
| FedNOVA | 93.51 | 36.50 | 86.47 | 56.17 | 97.66 | 79.92 |
| FedProto | 92.59 | 30.84 | 87.51 | 49.48 | 97.69 | 78.45 |
| MOON | 93.32 | 31.75 | 86.61 | 51.56 | 97.56 | 77.16 |
| FedType | 94.06 | 40.88 | 87.64 | 60.03 | 97.48 | 79.36 |
| AdaFGL | 85.44 | 36.13 | 88.22 | 56.91 | 92.88 | 79.69 |
| FedGTA | 85.43 | 35.40 | **88.79** | 56.17 | 92.86 | 80.02 |
| TRUST | **94.60** | **42.52** | 88.44 | **64.04** | **97.76** | **82.65** |

The results in Tab. 6 demonstrate that TRUST maintains superior performance even under extreme data heterogeneity, achieving the best performance on 5 out of the 6 evaluation metrics, with only the local accuracy on the Citeseer dataset showing slightly lower performance. These findings strongly validate the robustness and effectiveness of our approach.

# E Backward Distillation in FedType.

Backward distillation leverages knowledge distillation to transfer knowledge from proxy models to private models. However, since proxy models are smaller than private models, conventional distillation approaches may lead to performance degradation. To address this, FedType introduces an Uncertainty-based Behavior Imitation Learning method that selectively transfers high-confidence knowledge rather than complete logits. For every node $v_i$, we construct a prediction set $\mathcal{S}_i$, which guarantees inclusion of the true label with high probability (e.g., 95% confidence). To compute $\mathcal{S}_i$, we need to train a conformal model denoted as $cp$ using the validation dataset denoted as $D'$ following Split Conformal Prediction[36, 1]. After training the conformal model, FedType proposes a dynamic conformal prediction with Regularized Adaptive Prediction Sets (RAPS) to calculate the prediction set:

$$\mathcal{S}_i = \{y \mid u \cdot p_i^t(y) + \rho_i^t(y) + g(\Delta^t, \lambda) \cdot (o_i^t(y) - \kappa_{\text{reg}})^+ \leq \tau\},$$

$$\rho_i^t(y) = \sum p_i^t(y') \mathbb{1}_{\{p_i^t(y') > p_i^t(y)\}},$$

$$g(\Delta^t, \lambda) = \begin{cases} \lambda \cdot \Delta^t - \Delta^t + \lambda, & \text{if } \Delta^t < 0, \\ \lambda, & \text{otherwise,} \end{cases} \tag{19}$$

$$o_i^t(y) = |\{y' \mid \rho_i^t(y') > \rho_i^t(y)\}|,$$

where $p_i^t(y)$ denotes the probability of class y for data sample $v_i$ predicted by conformal model, $u$ is a randomization factor for prediction set construction. Furthermore, $\rho_i(y)$ denotes the total probability mass of labels more probable than $y$, $g(\Delta^t, \lambda)$ is a piecewise calibration function where $\Delta^t = A(p^t, D') - A(p^{t-1}, D')$ represents the accuracy difference between epoch $t$ and $t - 1$. Moreover, $o_i^t(y)$ denotes the label ranking of y based on predicted possiblity $\rho_i^t$, $\kappa_{\text{reg}}$ is a regularization hyper-parameter, and $(\cdot)^+$ is the positive part operator.

After constructing the prediction set, FedType computes the knowledge transfer weight $\eta_i$, which plays a crucial role in determining the amount of information transferred from the proxy model to the

private model. This weight is inversely proportional to the size of the prediction set $\mathcal{S}_i$. When $\mathcal{S}_i$ contains fewer labels, which indicates higher confidence in the prediction, we assign a larger $\eta_i$ to encourage the proxy model to transfer such confident knowledge to the private model. Conversely, larger prediction sets result in smaller transfer weights. Therefore, $\eta_i$ can be defined as:

$$\eta_i = \begin{cases} |\mathcal{S}_i \cap \mathcal{L}_i|/|\mathcal{S}_i \cup \mathcal{L}_i|, & \text{if } |\mathcal{S}_i| \geq |\mathcal{L}_i|, \\ |\mathcal{S}_i \cap \mathcal{L}_i|/|\mathcal{S}_i|, & \text{if } |\mathcal{S}_i| < |\mathcal{L}_i|, \end{cases} \tag{20}$$

where $\mathcal{S}_i$ and $\mathcal{L}_i$ are similarly computed using Equation 19 but with conformal model trained by proxy model and private model respectively. This formulation ensures adaptive knowledge transfer based on the proxy model's confidence level.

During backward distillation, we specifically enhance the probability alignment for labels within the prediction set through the following loss function:

$$L_{\text{backward}} = \sum_{i=1} \eta_i \sum_{y \in \mathcal{S}_i} \log\left(\frac{\exp(\mathbf{w}_i^t(v_i)[y])}{\Phi_i^t}\right),$$

$$\Phi_i^t = \sum_{y \in \mathcal{S}_i} \exp(\mathbf{w}_i^t(v_i)[y]) + \sum_{y' \in \hat{\mathcal{S}}_i} \exp(\mathbf{w}_i^t(v_i)[y']), \tag{21}$$

where $\hat{\mathcal{S}}_i = C - \mathcal{S}_i$ and represents labels with low confidence in the prediction.

## F    Complexity Analysis.

In this section, we present the complexity analysis of our proposed method TRUST. We begin with FedAvg as a baseline. Let $T$ denote the communication rounds. On the client side, each client executes $E$ epochs of local training per round with model parameter size $d$. When combined with GNN message passing, the computational complexity is $(L \cdot |\mathcal{E}| \cdot d)$ where $L$ and $|\mathcal{E}|$ represent the number of GNN layers and edges respectively. On the server side, model aggregation across $K$ clients yields $O(K \cdot d)$ complexity. Therefore, the total complexity is: $O(T \cdot (E \cdot L \cdot |\mathcal{E}| \cdot d + K \cdot d))$.

For FedType, knowledge distillation introduces an additional $O(N \cdot C)$ cost per epoch, where $N$ and $C$ are the number of nodes and classes respectively. The total complexity therefore becomes: $O(T \cdot (E \cdot (L \cdot |\mathcal{E}| \cdot d + N \cdot C) + K \cdot d))$.

For TRUST, it extends the knowledge distillation framework with three key components:

- **PCNS** first requires calculating node difficulties via neighborhood distribution entropy and prototype alignment. For neighborhood distribution entropy, it iterates through all edges ($O(|\mathcal{E}|)$). For prototype alignment, we pre-compute prototypes for all classes and calculate prototype similarity for every node ($O(N + C)$). Node sorting then adds $O(N \log N)$.
- **ACDM** involves only lightweight operations: temperature scaling $O(1)$ and cosine scheduling $O(1)$ per epoch, as indicated in Equation 13.
- **WDAD** leverages Sinkhorn algorithm to solve the loss function quickly. This reduces optimal transport complexity from $O(C!)$ to $O(k \cdot C^2)$, where $k$ is the iteration count (in our experiments this is set to 10).

Notably, WDAD's $O(k \cdot C^2)$ complexity remains manageable in practice since $C$ is bounded (e.g., C=7 for Cora and C=3 for Pubmed). Therefore, the total complexity is: $O(T \cdot (E \cdot (L \cdot |\mathcal{E}| \cdot d + N \cdot C + k \cdot C^2) + N \log N + K \cdot d))$.

As shown above, this represents only two additive terms compared to FedType: $O(N \log N)$ and $O(T \cdot E \cdot k \cdot C^2)$.

## G    Discussion on Limitations.

Although TRUST achieves significant success in addressing key challenges in model-heterogeneous Federated Graph Learning (FGL)—such as dynamic task difficulty adjustment, adaptive distillation signaling, and cross-class relational knowledge transfer—it still has some limitations. One notable challenge is the computational complexity involved in dynamically adjusting task difficulty and distillation strength, which could become a bottleneck in large-scale settings.

