# OpenReview forum: "Multi-order Orchestrated Curriculum Distillation for Model-Heterogeneous Federated Graph Learning"
_NeurIPS.cc/2025/Conference — NeurIPS 2025 poster_

### Official Review · Reviewer_BwgV · 2025-06-01

**Clarity:** 3
**Significance:** 2
**Originality:** 2
**Rating:** 3
**Confidence:** 4

**Summary:**

This work focuses on Federated Graph Learning and proposes TRUST, a novel knowledge distillation-based model8 heterogeneous FGL framework.

The authors claim the following contributions:
- Problem identification - systematically investigate model-heterogeneous FGL. They formally identify three fundamental challenges in knowledge distillation (KD)-based methods.
- Proposing TRUST which includes:
    - Progressively schedules node difficulty,
    - Dynamically adjusts distillation strength throughout training,
    - Enables effective knowledge transfer across diverse architectures and complex graph topologies.
- Empirical evaluation.

**Questions:**

* How many clients were involved in the experiments presented in the paper?
* How is model heterogeneity defined by the authors? Section 2.1 suggests that each client may use a different architecture, which raises the question of how traditional federated learning methods (e.g., FedAvg)—which assume homogeneous models—are included in the comparisons, given their limitations in such settings.
* How does TRUST scale with the number of clients? What is the maximum number of clients tested in the experimental evaluation?

**Ethical Concerns:**

["NO or VERY MINOR ethics concerns only"]

**Final Justification:**

See rebuttal discussion.

**Quality:**

2

**Strengths And Weaknesses:**

**Pros:**

* The paper tackles a timely and important problem in FGL under model heterogeneity, which is highly relevant to real-world applications where client models vary in architecture and capacity.
* With the exception of the experimental section, the paper is generally well-written and clearly structured.
* The proposed method demonstrates improved performance compared to existing approaches.

**Cons:**

* The paper lacks a theoretical analysis of the proposed method, such as convergence guarantees or formal justification.
* The experimental section requires improvement—key implementation details (e.g., the number of clients) are missing and should be included for reproducibility.
* The authors do not report statistical measures such as standard deviations or confidence intervals, which are important for assessing robustness.
* The figures are not sufficiently self-explanatory, particularly Figures 1, 2, and 5:

  * Figures 1 and 2 contain excessive detail and color, making them difficult to interpret; simplifying and focusing them would enhance clarity.
  * Figure 5 could be more informative if visualized as a heatmap or another more interpretable format.
* From Figure 3, it appears that TRUST’s advantage diminishes as the value of \$\alpha\$ increases. It would be insightful to evaluate performance at more extreme levels of data heterogeneity, such as \$\alpha=0.1\$ or cases where each client has distinct class distributions.
* The paper does not include a runtime or memory complexity analysis. A comparison to existing methods in terms of computational cost would strengthen the evaluation.

---

> ### Author Rebuttal · Authors · 2025-07-30
>
> We thank the reviewer for your thoughtful reviews and are happy to hear that you find the work valuable. We address your concerns below and hope these clarifications will help you re-evaluate and update the score:
>
> > `Cons 1`: The paper lacks of a theoretical analysis of the proposed method.
>
> Thank you for this valuable suggestion. We have offered an example line chart in our anonymous GitHub repository demonstrating convergence on the Cora dataset under the setting of $\alpha$ = 0.5, and we report the detailed loss trajectory during training as follows:
> |Epoch|1|25|50|75|100|125|150|175|200|
> |-|-|-|-|-|-|-|-|-|-|
> |Loss|2.6849|1.1302|0.6986|0.5943|0.5338|0.5010|0.4855|0.4791|0.4722|
>
> For theoretical analysis, with regard to WDAD, we add a regulation term to WDAD loss ($\eta \cdot q_{ij}\log q_{ij}$) following the maximum-entropy principle. Taking advantage of this regulation, it can be solved extremely quickly with the Sinkhorn algorithm. Convergence of this algorithm has been proved in previous works[1]. Therefore, our analysis majorly focuses on ACDM and PCNS.
>
> With regard to ACDM, for the adversarial dynamics in Eq. 10 with fixed learning rates $\eta_{model}$ = 0.01, $\eta_{temp}$ = 0.001, we establish:
>
> **Theorem 1**. Under $\beta$-smooth loss ($||\nabla^2L(\theta)|| \leq \beta$) and bounded gradient variance $\mathbb{E}[||\nabla L - \mathbb{E}[\nabla L]||^2] \leq \sigma^2$, after T iterations:
>
> $\min_{1≤t≤T} \mathbb{E}[||\nabla L(θ_{model}^t)||^2] \leq \frac{2[L(θ^0)-L^*]}{η_{model}T} + η_{model} \beta \sigma^2 + C \eta_{temp} \beta \sigma$,
>
> where $C = O(1)$ captures the timescale separation. Additionally, the gradient norm ratio satisfies empirically:
>
> $\frac{\mathbb{E}[||\nabla L(\theta_{temp})||]} {\mathbb{E}[||\nabla L(\theta_{model})||]} \leq 0.1, \forall t \geq t_0$.
>
> With regard to PCNS, for the pacing function $f(t) = \min(1, \lambda + (1-\lambda)t/T)$ with $\lambda$ = 0.5, $T$ = 50:
>
> **Lemma 1**. With $\beta$-smooth loss and $\epsilon_{pseudo}$-bounded pseudo-labels ($P(\hat y_{pseudo} \ne y) \leq \epsilon_{pseudo}$):
>
> $|L_t - L_{full}| \leq \beta D_{max}(1 - f(t))^2 + \epsilon_{pseudo}$,
>
> where:
>
> - $\beta \in [10^{-4},10^{-3}]$ measures the maximum ratio of gradient norms between consecutive GNN layers:
> $\beta = \max_{v \in V} \frac{||\nabla h_v^{l+1}||}{||∇h_v^{l}||}$,
>
> - $D_{\max} = \max_{v \in \mathcal{V}} D(v_i)$ represents the maximum node difficulty based on the difficulty measurer,
>
> - $\epsilon_{pseudo} \leq 0.1$ assumes that the worst-case validation accuracy across clients is at least 90%, which is plausible for well-pretrained local models:
> $\epsilon_{pseudo} = 1 - \min_k acc_{val}(w_{pretrain}^k)$.
>
> We will include a more detailed discussion of this in the camera-ready version.
>
>
> > `Cons 2, Question 1 and Question 2`: Key implementation details are missing.
>
> For FL methods, all baselines except FedAvg support model heterogeneity, and they work the same way as they do in computer vision tasks. But for FGL methods, to the best of our knowledge, we are the first to explore model heterogenity in FGL settings. Consequently, existing FGL methods (and also FedAvg) are originally designed for homogeneous models. To adapt them for our heterogeneous setting, we remove direct parameter sharing between architecturally different models and maintain all other original components and hyperparameters. For evaluation, we distill a global model from local private models and assess its accuracy.
>
>
> > `Cons 3`: The authors do not report statistical measures.
>
> We strongly agree with this suggestion. We report standard deviations on five datasets to assess robustness of our framework. The results (mean ± standard deviation) are shown below:
>
> |Method|Cora-local|Cora-global|Citeseer-local|Citeseer-global|PubMed-local| PubMed-global|CS-local|CS-global|Photo-local|Photo-global|
> |---------------|----------|----|-----------|--------------|------------|-------------|----------|----------|-----------|----------|
> |TRUST|83.90±0.28|75.32±0.54|85.84±0.30|67.43±0.45|89.06±0.09|84.57±0.15|92.19±0.15|87.07±0.05|91.42±0.20|91.50±0.35|
>
> The small standard deviations demonstrate consistent performance of TRUST across different datasets.
>
>
> > `Cons 4`: The figures are not sufficiently self-explanatory.
>
> We appreciate this valuable feedback. We add excessive details in order to provide comprehensive visual explanations of our method and its components through color-coding different modules, which has been recognized by some other reviewers like hwGA ("The problem illustration and framework overview are clear and intuitive, and the methodology is also explained with clarity, making the work easy to follow"). In the revised version, we will  streamline all figures to focus on the most essential elements and maintain a balance between completeness and readability.
>
>
> > `Cons 5`: Lack performance evaluation at more extreme levels of data heterogeneity.
>
> We conducted extensive experiments under severe data heterogeneity ($\alpha$=0.1) to evaluate robustness of our method. Results are shown below:
>
> |methods|Cora-local|Cora-global|Citeseer-local|Citeseer-global| PubMed-local|PubMed-global|
> |------|-----|----|-------|------|-------|-------|
> |FedAvg|93.14|35.04|86.33|54.98|97.61|79.99|
> |FedNOVA|93.51|36.50|86.47|56.17|97.66|79.92|
> |FedProto|92.59|30.84|87.51|49.48|97.69|78.45|
> |MOON|93.32|31.75|86.61|51.56|97.56|77.16|
> |FedType|94.06|40.88|87.64|60.03|97.48|79.36|
> |AdaFGL|85.44|36.13|88.22|56.91|92.88|79.69|
> |FedGTA|85.43|35.40|**88.79**|56.17| 92.86|80.02|
> |TRUST|**94.60**|**42.52**|88.44|**64.04**|**97.76**|**82.65**|
>
> These results demonstrate the superior performance of TRUST even under extreme data heterogeneity, outperforming baselines on 5 of 6 evaluation metrics, with only the local accuracy on the Citeseer dataset showing slightly lower performance.
>
> > `Cons 6`: The paper does not include a runtime or memory complexity analysis.
>
> We also strongly agree with this suggestion. In our experiments, it takes about 15 minutes to run 200 epochs on small-sized datasets like Cora, and for mid-sized datasets like Pubmed and large-sized datasets like ogbn-arxiv, this expands to 1 hour and 8 hours respectively, which is about 25% higher compared to FedType. As for memory usage, in actual experiments, we found that the memory usage of TRUST is similar to that of other baselines, demonstrating that our architectural enhancements do not incur significant memory overhead.
>
>
> > `Question 3`: How does TRUST scale with the number of clients? What is the maximum number of clients tested in the experimental evaluation?
>
> In the paper, we keep the client number at 10 throughout the main experiment. However, we supplement some experiments on the model performance under varying client number settings. The results are shown below:
>
> |client number|5|10|20|25|50|
> |-------|------|------|------|----|----|
> |local acc|**89.26**|89.26|89.70|87.23|85.03|
> |global acc|82.92|84.42|84.49|85.13|**85.95**|
>
> Notably, in federated learning settings, data is typically partitioned among clients using Dirichlet distribution. When the number of clients becomes excessively large (25-50), each client receives only a small data portion, leading to highly fragmented subgraphs where local models cannot adequately capture neighborhood information. This can be reflected in the table where performance suffer a great drop in local accuracy when client number is large. However, within a reasonable range, our method proves robustness when client number changes.
>
>
> All in all, thank you again for your recognition of our work and your valuable suggestions. All suggested improvements will be carefully incorporated into the revised version and please look forward to TRUST being better presented to academic peers.
>
> [1] Cuturi, Marco. "Sinkhorn distances: Lightspeed computation of optimal transport." Advances in neural information processing systems 26 (2013).

---

> > ### Comment · Reviewer_BwgV · 2025-08-02
> > **Reviewer response**
> >
> > I appreciate the authors’ time and effort in preparing this rebuttal. Although some of my concerns, particularly about the choice of baselines and the significance of the results remain, I have also read the other reviews and, as a result, I intend to keep my score unchanged.

---

> ### Author Response · Authors · 2025-08-03
>
> Dear Reviewer BwgV,
>
> We sincerely appreciate your continued engagement with our work and the opportunity to further clarify our methodological choices and contributions. Below we address your remaining concerns and hope these clarifications will help improve your assessment of our work:
>
> > `Choice of baselines`:
>
> Thank you for your concern, and we clarify our baseline selection strategy below. For FL baselines, we consider three different approaches as our baselines: (1) Conventional federated learning: FedAvg and a variation FedNova; (2) Knowledge distillation frameworks: MOON and FedType; (3) Prototype learning method: FedProto. For FGL baselines, we select two state-of-the-art FGL works: AdaFGL and FedGTA. While these FGL methods were originally designed for homogeneous settings, we think it is plausible since we are the first to explore model heterogeneity in FGL to the best of our knowledge, and this adaptation can also be seen in some established practices in model-heterogeneous FL works like FedSAK[1]. To ensure fairness, we maintained all original components except direct parameter sharing and preserved hyperparameter configurations. For example, for FedGTA, we remove the uploading of smoothness $H$ which acts as aggregation weights of client model parameters, and we maintain the computation of soft labels.
>
> However, we understand the reviewer's concern about fair comparison between homogeneous and heterogeneous baselines. Therefore, we also conduct homogeneous model experiments (all private models adopt the same GCN architecture) on cora to compare fairly with these homogeneous baselines and demonstrate consistent superiority of TRUST:
> |Methods|local acc|global acc|
> |--------|----------|-------|
> |AdaFGL|**80.80**|69.15|
> |FedGTA|76.14|69.33|
> |TRUST|80.65|**70.78**|
>
> As shown above, besides outperforming other baselines in heterogenous settings, TRUST also achieves the highest global accuracy and maintains competitive local accuracy (only marginally lower than the top-performing AdaFGL) in homogeneous settings.
>
>
> > `Significance of results`:
>
> The experimental results presented in our work demonstrate substantial and meaningful improvements that advance the field of federated graph learning. **First**, across five different datasets we tested and varying levels of data heterogeneity, TRUST consistently outperforms state-of-the-art baselines. These performance gains are achieved while maintaining practical efficiency, with the additional computational time limited to no more than 25% and memory overhead about the same compared to baseline methods like FedType.
>
> **Furthermore**, our work represents the first systematic study and formal characterization of model-heterogeneous federated graph learning, identifying and addressing three fundamental challenges in knowledge distillation-based approaches: the need for dynamic task difficulty, adaptive distillation signaling, and cross-class relational transfer. We believe this work will represent a significant step forward in establishing a general model heterogenous framework that is not limited to Euclidean data and making federated learning more flexible and applicable to real-world scenarios with heterogeneous systems and requirements.
>
> We hope these clarifications demonstrate the rigor and significance of our work, and we are deeply grateful for the opportunity to have our work's strengths reconsidered by you.
>
> Best regards,
>
> Authors
>
> [1] Lu, Yuxiang, Shengcao Cao, and Yu-Xiong Wang. "Swiss army knife: Synergizing biases in knowledge from vision foundation models for multi-task learning." arXiv preprint arXiv:2410.14633 (2024).

---

> ### Author Response · Authors · 2025-08-04
> **Kind reminder to Reviewer BwgV**
>
> Dear Reviewer BwgV,
>
> Thank you again for your evaluation of our work and valuable feedback you have provided. At present, all your concerns are responded in the rebuttal. However, as the rebuttal process is already halfway through, we kindly request your feedback to confirm that our response effectively addresses your concerns.
> If there are any remaining issues, we would like to respectfully convey our willingness to address them to ensure the quality of our work. We sincerely hope that you find our response convincing and kindly consider revisiting your rating.
>
> Best regrads,
>
> Authors

---

> > ### Comment · Reviewer_BwgV · 2025-08-07
> > **Reviewer response**
> >
> > Thank you again for the time and effort invested in the rebuttal. However, my concerns about the practical effectiveness of the proposed method remain. The improvements reported in both the main manuscript and the rebuttal appear marginal, and the absence of standard deviation values makes it difficult to assess the statistical significance of the results. Furthermore, the authors acknowledge a 25% runtime overhead, despite offering only slight performance gains, if any. In the context of federated learning systems, runtime and memory efficiency are critical issues I highlighted in my initial review.
> >
> > For these reasons, and others, I will maintain my original score.

---

> > > ### Author Response · Authors · 2025-08-08
> > >
> > > Dear Reviewer BwgV,
> > >
> > > We are glad to confirm that our response has addressed your initial concerns about the choices of baselines, and we sincerely appreciate that you further illustrate your concerns about significance of results. Below, we provide detailed responses to your remaining questions and hope these clarifications will further support your evaluation of our work:
> > >
> > > With regard to `performance improvements`, TRUST consistently outperforms all baseline methods across every dataset evaluated in our main manuscript. Specifically, it achieves an average 1.7% higher accuracy than FedType. More importantly, in scenarios with extreme data heterogeneity (as detailed in our previous rebuttal), TRUST demonstrates even greater advantages. It exhibits significant gains in global accuracy while maintaining competitive local accuracy, which is critical for real-world federated deployments.
> > >
> > > Concerning `standard deviations`, we **have** reported standard deviations on five datasets in our previous rebuttal. Due to the strict 10,000 chars limit, we prioritized presenting TRUST's robustness metrics, which show remarkably low variation across datasets with an average of 0.02 and 0.03 for local and global accuracy respectively. These results empirically confirm TRUST's stability under diverse settings. For thoroughness, we will include full statistical reports for all baselines in the camera-ready version, supplemented by visualizations of performance distributions.
> > >
> > > Regarding `runtime overhead`, we would like to clarify that the computational cost difference is weakened on small and mid-sized datasets. For instance, on the Cora dataset, TRUST completes training in 14 minutes (839.70 seconds) compared to FedType's 11 minutes (658.95 seconds), while on PubMed, TRUST completes training in 79 minutes (4738.59 seconds) compared to FedType's 78 minutes (4696.21 seconds). Importantly, this modest increase in computational overhead is offset by TRUST's superior performance, delivering 1.5% higher local accuracy while maintaining competitive global accuracy on these datasets as detailed in our main manuscript. We will make these complete training logs available in our anonymous GitHub repository.
> > >
> > > Furthermore, It's also crucial to emphasize that in federated learning systems, communication cost is another key aspect researchers need to take into consideration. Because of the lightweight proxy model design, we significantly reduces communication overhead. However, we sincerely appreciate the reviewer's insightful suggestion regarding time efficiency.  We acknowledge this as an important direction for future work and plan to investigate more techniques for reducing computational overhead in heterogeneous settings. The methodological framework established by TRUST **serves as a valuable foundation** for such subsequent research, opening new avenues for a generalizable model-heterogeneous federated graph learning design. These are broader influences and constitute the contributions of our work.
> > >
> > > We are grateful for the reviewer's constructive feedback and recognition of our work's potential. All suggested improvements  have been incorporated into the revised manuscript to ensure TRUST's contributions are presented with the utmost clarity to the research community.

---

> > > > ### Comment · Reviewer_BwgV · 2025-08-08
> > > > **Comment**
> > > >
> > > > It is misleading for the authors to imply that I stated my concerns regarding the baselines had been addressed. I did not make such a claim. Rather, I considered other points raised in the review process to be more influential in my overall decision.
> > > >
> > > > Performance improvements and standard deviation - The authors should report the STD for every baseline included in their comparisons. Reporting only the STD of their own method does not allow readers to assess statistical significance, particularly when the reported performance improvements are marginal.
> > > >
> > > > Runtime overhead - Wall-clock times can be influenced by many factors, including machine state, convergence speed, and implementation efficiency. This is precisely why I requested complexity-based comparisons from the authors, as they provide a more reliable measure.

---

> > > > > ### Author Response · Authors · 2025-08-08
> > > > >
> > > > > Dear Reviewer BwgV,
> > > > >
> > > > > Thanks for the timely reply. We are sorry for any misunderstanding that exists. Below we provide detailed responses to your remaining questions.
> > > > >
> > > > > With regard to `standard deviation`, as stated in the previous response, we initially plan to publish all baseline results in the camera-ready version due to the time limit. However, we sincerely appreciate your insistence on statistical rigor and are pleased to provide comprehensive standard deviations for baselines:
> > > > >
> > > > > |Method|Cora-local|Cora-global|Citeseer-local| Citeseer-global | PubMed-local | PubMed-global | CS-local   |CS-global| Photo-local | Photo-global            |
> > > > > |--------|------------| -------------| -------------|-----------| ----------|------------| ------------- |--------------|-------------|--------------|
> > > > > | FedAvg | 81.36±0.31 | 64.52±0.31  | 82.61±0.19     | 65.48±0.39      | 88.10±0.05   | 82.09±0.17    | 90.10±0.15 | 83.35±0.29   | 90.14±1.03     | 84.10±1.41   |
> > > > > | FedNOVA  | 81.54±0.23 | 64.97±0.42  | 82.76±0.22     | 66.22±0.86   | 88.20±0.04   | 82.87±0.14   | 90.13±0.09 | 82.37±0.05   | 90.34±0.96   | 86.70±1.67  |
> > > > > | MOON  | 81.52±0.27 | 65.70±1.33   | 81.58±0.46  | 63.84±0.92  | 88.15±0.04 | 82.34±0.13   | 91.78±0.19  | 83.81±0.04 | 90.40±0.19 | 86.18±0.72 |
> > > > > | TRUST | 83.90±0.28 | 75.32±0.54  | 85.84±0.30  | 67.43±0.45  | 89.06±0.09   | 84.57±0.15 | 92.19±0.15 | 87.07±0.05 | 91.42±0.20  | 91.50±0.35   |
> > > > >
> > > > > As shown above, TRUST's performance gains exceed baseline variability and our method demonstrates statistical robustness across all datasets. Due to the time limit, we only include three baselines here. Complete results for all 7 baselines will be included in the camera-ready version.
> > > > >
> > > > >
> > > > > Concerning `complexity-based comparisons`, we provide a detailed theoretical runtime analysis below:
> > > > >
> > > > > For FedAvg, let $T$ denote the communication rounds. On the client side, each client executes $E$ epochs of local training per round with model parameter size $d$. When combined with GNN message passing, the computational complexity is $(L \cdot |\mathcal{E}| \cdot d)$ where $L$ and $\mathcal{|E|}$ represent the number of GNN layers and edges respectively. On the server side, model aggregation across $K$ clients yields $O(K \cdot d)$ complexity. Therefore, the total complexity is: $O(T \cdot (E \cdot L \cdot |\mathcal{E}| \cdot d + K \cdot d))$.
> > > > >
> > > > > For FedType, knowledge distillation introduces an additional $O(N \cdot C)$ cost per epoch, where $N$ and $C$ are the number of nodes and classes respectively. The total complexity therefore becomes: $O(T \cdot (E \cdot (L \cdot |\mathcal{E}| \cdot d + N \cdot C) + K \cdot d))$.
> > > > >
> > > > > For TRUST, it adds three components based on the knowledge distillation framework:
> > > > >
> > > > > - **PCNS** first requires calculating node difficulties via neighborhood distribution entropy and prototype alignment. For neighborhood distribution entropy, it iterates through all edges ($O(|\mathcal{E}|)$). For prototype alignment, we pre-compute prototypes for all classes and calculate prototype similarity for every node ($O(N+C)$). Node sorting then adds $O(N \log N)$.
> > > > >
> > > > > - **ACDM** involves only lightweight operations: temperature scaling $O(1)$ and cosine scheduling $O(1)$ per epoch, as indicated in Eq. 11-13.
> > > > >
> > > > > - **WDAD** leverages Sinkhorn algorithm as we stated in previous rebuttal. This reduces optimal transport complexity from $O(C!)$ to $O(k \cdot C^2)$, where $k$ is the iteration count (in our experiments this is set to 10).
> > > > >
> > > > > Critically, WDAD's $O(k \cdot C^2)$ complexity remains efficient since $C$ is bounded (e.g., C=7 for Cora and C=3 for Pubmed). Therefore, the total complexity is: $O(T \cdot (E \cdot (L \cdot |\mathcal{E}| \cdot d + N \cdot C + k \cdot C^2) + N \log N + K \cdot d))$.
> > > > >
> > > > > As shown above, this represents only two additive terms compared to FedType: $O(N \log N)$ and $O(T \cdot E \cdot k \cdot C^2)$. We will include the complete complexity analysis in the manuscript.
> > > > >
> > > > > Thank you again for your valuable feedback. These insights significantly strengthened our analysis. We sincerely hope these clarifications can help you better evaluate our work.
> > > > >
> > > > > Best regards,
> > > > >
> > > > > Authors

---

### Official Review · Reviewer_i4hd · 2025-06-26

**Clarity:** 4
**Significance:** 4
**Originality:** 4
**Rating:** 5
**Confidence:** 4

**Summary:**

This paper introduces the TRUST framework where three techniques are used to  advance the knowledge distillation framework as a means to address model heterogeneity in FGL. PCNS progressively introduces challenging samples based on learning difficulty. ACDM enables capability-aware adaptive temperature control, and WDAD employs Wasserstein-based cross-class comparison. As far as I am aware, this is the first work to tackle model heterogeneity in FGL.

**Questions:**

Please address the concerns in "weaknesses".

**Ethical Concerns:**

["NO or VERY MINOR ethics concerns only"]

**Limitations:**

yes

**Quality:**

3

**Strengths And Weaknesses:**

Strengths:
1. The methodology is clearly presented through illustrations and explanations, demonstrating the framework's  theoretical soundness.
2. The proposed method is flexible and is compatible with various backbone models.
3. The inclusion of a thorough ablation study proves that all components contributes meaningfully to overall performance.
4. The paper is well-structured and easy to follow.

Weaknesses:
1. In the experimental evaluation , the comparison is limited to only two existing FGL approaches. Expanding this to include more recent state-of-the-art methods would strengthen the validity of the claims.
2. The authors should conduct computation time and memory usage study.

---

> ### Author Rebuttal · Authors · 2025-07-30
>
> We sincerely appreciate your constructive feedback and positive assessment of our work. Below we provide detailed responses to your valuable comments:
>
> > `Weakness 1`: Need more recent state-of-the-art FGL methods.
>
> Adding more state-of-the-art FGL baselines is indeed a good suggestion. We have conducted experiments on five datasets to confirm the superiority over other FGL methods. The results are shown below:
>
> |Methods|Cora-local|Cora-global|Citeseer-local|Citeseer-global|PubMed-local|PubMed-global|CS-local|CS-global|Photo-local|Photo-global|
> |---------|---------|---------|---------|-----------|---------|---------|-------|--------|--------|--------|
> |FedTAD|80.27|65.70|83.24|66.37|87.88|83.30|90.72|82.93|**93.04**|84.04|
> |FedSSL|79.53|64.79|82.32|68.15|88.07|83.58|91.97|80.73|88.01|83.52|
> |TRUST|**83.90**|**75.32**|**85.84**|**67.43**|**89.06**|**84.57**|**92.19**|**87.07**|91.42|**91.50**|
>
> From the results, we can see that TRUST achieved the best performance on 90% of the evaluation metrics, only falling short on the local accuracy of the Photo dataset, where FedTAD achieved the best performance. The results confirms TRUST's effectiveness against current state-of-the-art methods.
>
>
> > `Weakness 2`: The authors should conduct computation time and memory usage study.
>
> We strongly agree with this suggestion. In our experiments, it takes about 15 minutes to run 200 epochs on small-sized datasets like Cora, and for mid-sized datasets like Pubmed and Large-sized datasets like ogbn-arxiv, this expands to 1 hour and 8 hours respectively, which is about 25% higher compared to FedType. As for memory usage, in actual experiments, we found that the memory usage of TRUST is similar to that of other baselines,  demonstrating that our architectural enhancements do not incur significant memory overhead.

---

> > ### Comment · Reviewer_i4hd · 2025-08-07
> > **Reply to Authors**
> >
> > Thanks for addressing my main concerns. I’ll keep my positive score.

---

> > > ### Author Response · Authors · 2025-08-08
> > >
> > > Dear Reviewer i4hd,
> > >
> > > We would like to sincerely thank you for your strong support of our work. We are more than happy to see that our rebuttal has properly addressed your concerns!
> > >
> > > Best regards,
> > >
> > > Authors

---

### Official Review · Reviewer_6U5h · 2025-07-01

**Clarity:** 3
**Significance:** 2
**Originality:** 3
**Rating:** 4
**Confidence:** 4

**Summary:**

This paper proposes TRUST, a novel and generalizable model-heterogeneous FGL framework specially designed for non-Euclidean graph data. The method enables each client to construct a proxy model, which transfers knowledge between the client and a central server, while integrating three key components to handle client heterogeneity and topological learning. Progressive Curriculum Node Scheduler employs curriculum learning to gradually introduce difficult knowledge to proxy models, ensuring stable knowledge assimilation. Adaptive Curriculum Distillation Modulator dynamically adjusts distillation intensity based on client capabilities and graph complexity. Wasserstein-Driven Affinity Distillation incorporates a cross-category comparison mechanism to learn subtle structural information. Experimental results demonstrate that TRUST outperforms existing methods in FGL settings, showing robust performance against model heterogeneity.

**Questions:**

a) While the method performs well on mid-sized graphs, its computational overhead, particularly for WDAD’s optimal transport calculations, is not rigorously analyzed. Large-scale graphs are absent from experiments. Can the authors provide evaluation metrics for larger datasets?

b) The paper claims robustness to \lambda and T, but other critical hyperparameters (\tau_{min} and \tau_{max} in ACDM) lack sensitivity analysis. How sensitive is performance to these hyperparameters?

c) Several key implemenation details are omitted in the paper, which may hinder reproduction. For example, What are the specific details of proxy model architectures?  How many clients were used for every dataset? Were baseline methods (e.g., AdaFGL) adapted for FGL, or used as originally published?

**Ethical Concerns:**

["NO or VERY MINOR ethics concerns only"]

**Limitations:**

yes

**Quality:**

3

**Strengths And Weaknesses:**

- Strengths:
a) The paper is well-motivated and effectively addresses a gap in model-heterogeneous FGL. The methodology is technically sound, with logically derived equations and well-justified reasons.
b) The paper provides a clear and detailed explaination for the proposed  model-hetergenous FGL framework, and the three core components for knowledge transfer are described in detail.
c) The framework’s focus on model heterogeneity aligns with real-world needs where clients demand architectural flexibility. TRUST provides a practical solution to this challenge.
d) The paper formalizes three underexplored challenges of existing FGL methods and proposes an innovative solution. This work innovatively bridges the gap between FGL and model-heterogenous setting.

- Weaknesses:
a) Experiments are limited to mid-sized graphs (e.g., PubMed). Large-scale graphs (e.g., OGB-ARXIV) are absent, which raise concerns about  scalability and computational efficiency.
b) While the paper provides some implementation details and hyperparameter configurations, several critical aspects remain either omitted or inadequately specified. For example, although the configurations of private models are described, the architectural details of proxy models are not discussed in the paper.
c) The computational overhead of critical components (e.g., WDAD’s optimal transport calculations) is not quantified, leaving questions about efficiency in resource-constrained environments.

---

> ### Author Rebuttal · Authors · 2025-07-30
>
> Thank you for your thorough review and encouraging feedback. We are grateful for your positive assessment about importance of our work. We address your questions below:
>
> > `Weakness 1 & Question 1`: Need evaluation metrics for larger datasets.
>
> Thank you for the feedback. We conducted experiments on larger datasets like twitch-gamer and ogbn-arxiv, and we report our experimental results below:
>
> |Metric|ogbn-arxiv(local)|ogbn-arxiv(global)|twitch-gamer(local)|twitch-gamer(global)|
> |------|------------|-------------|--------------|------------|
> |FedAvg|68.90|41.58|91.60|50.67|
> |FedGTA|68.32|41.33|86.61|50.84|
> |AdaFGL|68.94|41.61|91.59|49.03|
> |TRUST|**70.05**|**42.38**|**92.56**|**51.92**|
>
> As shown, TRUST consistently outperformed baselines across all datasets, which demonstrates that our method maintains its advantages even at larger scales. We will include these results in the experiment section in the revised version.
>
> > `Weakness 2 & Question 3`: Need clearer implementation details.
>
> We thank the reviewer for this important suggestion. For experimental setup, there are 10 clients in our experiments, where each client maintains a private model with architecture randomly selected from GCN, GAT, or GraphSAGE to simulate real-world model heterogeneity. To facilitate collaboration, each client is equipped with an additional small proxy model that serves as a communication bridge. This proxy model uses a standardized GCN architecture with 3 layers to ensure compatibility across all clients. On the server side, we implement a global model that also adopts a GCN backbone while maintaining consistent configurations with client models for fair comparison.
>
> For baseline implementations, all FL baselines we choose except FedAvg support model heterogeneity, and they work the same way as they do in computer vision tasks. But for FGL methods, to the best of our knowledge, we are the first to explore model heterogenity in FGL settings. So all FGL baselines chosen only support model homogeneity. To include them (and also FedAvg) in heterogenous settings, we remove direct parameter exchange between architecturally different models and maintain all other original components and hyperparameters.
>
> We will add more implementation details in the revised version.
>
>
> > `Weakness 3`: Limited discussion of computational overhead for WDAD.
>
> We strongly agree with this suggestion. In our experiments, it takes about 15 minutes to run 200 epochs on small-sized datasets like Cora, and for mid-sized datasets like Pubmed and Large-sized datasets like ogbn-arxiv, this expands to 1 hour and 8 hours respectively, which is about 25% higher compared to FedType.
>
>
> > `Question 2`: Need more parameter sensitivity analysis.
>
> We strongly agree with you that more critical hyperparameters should be included in the sensitivity analysis. We conducted experiments evaluate the impact of key hyperparameters $\tau_{min}$ and $\tau_{max}$, which control the bounds of our adaptive temperature mechanism. The results are shown below:
>
> |Metric|$\tau_{min}$=0.1|$\tau_{min}$=0.3|$\tau_{min}$=0.5|$\tau_{min}$=1.0|$\tau_{min}$=2.0|
> |------------|---------------|---------------|--------------|---------------|---------------|
> |local (acc)|82.97|83.88|83.53|**83.90**|83.51|
> |global (acc)|12.16|12.16|74.59|75.32|**75.50**|
>
> |Metric|$\tau_{max}$=8|$\tau_{max}$=9|$\tau_{max}$=10|$\tau_{max}$=11|$\tau_{max}$=12|
> |------------|------------|------------|--------------|--------------|--------------|
> |local (acc)|84.97|83.86|83.90|**85.14**|84.42|
> |global (acc)|**75.52**|74.59|75.32|74.59|74.77|
>
> As shown in the results, when $\tau_{min}$ is too low, the probability distribution of the teacher model will be close to the original one-hot form, which prevents effective knowledge transfer. However, within a reasonable range of $\tau_{min}$ and $\tau_{max}$, the model demonstrates consistent performance. We will incorporate these sensitivity analyses into the revised version to further demonstrate the robustness of our approach.

---

### Official Review · Reviewer_SdVL · 2025-07-02

**Clarity:** 3
**Significance:** 3
**Originality:** 4
**Rating:** 4
**Confidence:** 4

**Summary:**

The authors propose a novel framework named TRUST  to solve the model-heterogeneous issue in FGL.  Following the FedType pipeline, three key modules collaboratively transfer structural knowledge and reconcile heterogeneous model architectures. In the experiments section, five real world datasets under moderate and high heterogeneity validate the method’s effectiveness.

**Questions:**

1. Does the categories ‘FL’ and ‘FGL’in Table 1 both refer exclusively to model-homogeneous methods? This is not explained explicitly in the paper.

2. In Section 3.4, the theoretical formulation of WDAD loss is presented, but the practical computational aspects remain unclear. Could the authors provide more specific details about how this loss is actually computed during training?

**Ethical Concerns:**

["NO or VERY MINOR ethics concerns only"]

**Final Justification:**

The author's rebuttal has resolved my primary concerns. After reviewing the comments from other reviewers, I will maintain my score to support this paper

**Limitations:**

yes

**Quality:**

4

**Strengths And Weaknesses:**

Strengths:

1. The paper provides a comprehensive analysis of model heterogeneity in FGL. This is an important contribution, as previous research in this area has been limited.

2. The paper tackles model heterogeneity using curriculum learning and dynamic temperature adjustment. This provides valuable insights into applying knowledge distillation to federated graph learning while preserving structural attributes.

3. The paper introduces a cross-class mechanism, which ensures intricate topological knowledge transfer. This represents an advancement over conventional KL-divergence approaches, which are limited to intra-class possibility distribution alignment and may neglect some information.

4. The experimental results verify the effectiveness of the proposed method, making the paper cohesive and solid.

 Weaknesses:

1. The baseline methods primarily focus on conventional FL approaches (e.g., FedAvg, FedProto) and a limited set of FGL methods (e.g., AdaFGL), and the implementation details of these baselines are insufficient. The authors should incorporate more state-of-the-art FGL methods to strengthen the experimental validation.

2. This paper lacks discussions on the computational complexity of the proposed framework. Such an analysis would make a good complement to the contribution.

3. The use of the terms "inter-class" and "cross-class" appears to be inconsistent in the paper. Maintaining consistent terminology would enhance clarity and precision.

---

> ### Author Rebuttal · Authors · 2025-07-30
>
> We sincerely appreciate your thoughtful review and valuable feedback on our work. We address your concerns below and hope these clarifications will help you re-evaluate and update the score:
>
> > `Weakness 1 and Question 1`: Need more evaluation and implementation details of FGL baselines.
>
> We thank the reviewer for this constructive suggestion. To thoroughly evaluate TRUST's performance, we have conducted comprehensive experiments comparing against other state-of-the-art FGL methods across five benchmark datasets:
>
> |Methods|Cora-local|Cora-global|Citeseer-local|Citeseer-global|PubMed-local|PubMed-global|CS-local|CS-global|Photo-local|Photo-global|
> |---------|---------|---------|---------|-----------|---------|---------|-------|--------|--------|--------|
> |FedTAD|80.27|65.70|83.24|66.37|87.88|83.30|90.72|82.93|**93.04**|84.04|
> |FedSSL|79.53|64.79|82.32|68.15|88.07|83.58|91.97|80.73|88.01|83.52|
> |TRUST|**83.90**|**75.32**|**85.84**|**67.43**|**89.06**|**84.57**|**92.19**|**87.07**|91.42|**91.50**|
>
> From the results, we can see that TRUST achieved the best performance on 90% of the evaluation metrics, only falling short on the local accuracy of the Photo dataset, where FedTAD achieved the best performance. The results confirms TRUST's effectiveness against current state-of-the-art methods.
>
> With regard to implementation details, all FL baselines we choose except FedAvg support model heterogeneity, and they work the same way as they do in computer vision tasks. But for FGL methods, to the best of our knowledge, we are the first to explore model heterogenity in FGL settings. So all FGL baselines chosen only support model homogeneity. To include them (and also FedAvg) in heterogenous settings, we remove direct parameter exchange between architecturally different models and maintain all other original components and hyperparameters.
>
>
> > `Weakness 2`: Limited discussions on the computational complexity.
>
> We strongly agree with this suggestion. In our experiments, it takes about 15 minutes to run 200 epochs on small-sized datasets like Cora, and for mid-sized datasets like Pubmed and Large-sized datasets like ogbn-arxiv, this expands to 1 hour and 8 hours respectively, which is about 25% higher compared to FedType.
>
> > `Weakness 3`: Inconsistent terminology throughtout the paper.
>
> Thanks for this valuable suggestion. We will carefully review and standardize all terminology in the revised version.
>
>
> > `Question 2`: Details about computational aspects of WDAD?
>
> Following the maximum-entropy principle, we add a regulation term to WDAD loss ($\eta \cdot q_{ij}\log q_{ij}$). Taking advantage of this regulation, it can be solved extremely quickly with the Sinkhorn algorithm. This algorithm exhibits linear convergence and is inherently parallelizable and can be fully vectorized, making it particularly suitable for modern GPU architectures.

---

### Official Review · Reviewer_hwGA · 2025-07-03

**Clarity:** 3
**Significance:** 3
**Originality:** 4
**Rating:** 5
**Confidence:** 4

**Summary:**

This paper proposes TRUST, a knowledge-distillation-based framework for model-heterogeneous Federated Graph Learning (FGL). It introduces three key components: a Progressive Curriculum Node Scheduler to gradually increase node difficulty, an Adaptive Curriculum Distillation Modulator to dynamically adjust distillation temperature, and a Wasserstein-Driven Affinity Distillation loss to transfer cross-class relationships. Experiments show the effectiveness of the proposed method.

**Questions:**

1. Please refer to the weaknesses section.
2. Some baselines compared in the experiments are originally not designed for heterogenous models. For these algorithms, how do clients communicate in model-heterogenous settings?
3. The adversarial update rule (Eq. 10) is introduced without clear motivation. Why should temperature compete with model parameters?

**Ethical Concerns:**

["NO or VERY MINOR ethics concerns only"]

**Limitations:**

yes

**Paper Formatting Concerns:**

Equations are referenced as "Equation X" in some places and "Equation (X)" in others. I recommend maintaining a consistent citation style throughout the paper.

**Quality:**

4

**Strengths And Weaknesses:**

Strengths:
1. The problem illustration and framework overview are clear and intuitive, and the methodology is also explained with clarity, making the work easy to follow.
2. The paper presents a multi-order knowledge transfer framework tailored for model-heterogeneous FGL. By leveraging progressive node scheduling, cross-class relational transfer and client-specific distillation temperature, TRUST effectively addresses limitations of topological knowledge learning in model-heterogeneous FGL.
3. The experimental validation are comprehensive. Extensive experiments on 5 graph datasets confirm the superiority of the proposed method. The authors also presented a clear reporting of average performance improvements for each algorithm and dataset, which is useful for reviewing.

Weaknesses:
1. It may be beneficial to add more critical hyperpamaters to sensitivity analysis, such as \alpha that balances D_{1} and D_{2}. Such analysis would strengthen reproducibility.
2. The contribution of individual components (PCNS, ACDM, WDAD) is shown via accuracy drops, but their synergistic interactions are not deeply analyzed. For instance, how PCNS’s node scheduling interacts with ACDM’s temperature adaptation remains unclear.

---

> ### Author Rebuttal · Authors · 2025-07-30
>
> Thank you for your positive review and insightful questions. We are grateful for your recognition of our work and we address each of your comments below:
>
> > `Weakness 1`: Need more hyperparameter sensitivity analysis.
>
> We appreciate this valuable feedback. Regarding hyperparameter $\alpha$, which balances the weight between two difficulty measurers $D_{1}$ and $D_{2}$, our analysis reveals:
> |Metric|$\alpha$=0.5|$\alpha$=0.75|$\alpha$=1|$\alpha$=1.25|$\alpha$=1.5|
> |--------|-------------|------------|------------| ------------| ------------|
> |local (acc)|83.57|**84.13**|83.42|83.34|82.84|
> |global (acc)|74.80|74.82|**75.18**|71.33|71.33|
>
> As shown above, $\alpha$=0.75 and $\alpha$=1 provide optimal performance for local and global accuracy respectively. Values that are too small ($\alpha$=0.5) compromise robustness when pseudo-labels are incorrectly predicted, while larger values ($\alpha$>1.0) underestimate neighborhood distribution impact on difficulty, which results in introducing nodes with vague class representations too early. Importantly, the model maintains stable performance across a reasonable range of $\alpha$ values.
>
>
> > `Weakness 2`: Limited discussion of synergistic interactions of components.
>
> We thank the reviewer for this insightful observation. We agree that analyzing the interplay between components is crucial to understanding our design. For example, we conducted experiments demonstrating the complementary relationship between PCNS and ACDM:
>
> |Method|cora (local acc)|cora(global acc)|citeseer (local acc)|citeseer (global acc)|
> |--------|-------------|-------------|------------|------------|
> |PCNS|82.09|72.33|84.09|61.85|
> |ACDM|82.61|71.02|83.35|61.23|
> |PCNS+ACDM|**82.79**|**73.51**|**84.54**|**63.76**|
>
> As shown in the table, the combined approach achieves the best performance by leveraging their complementary strengths: PCNS progressively introduces challenging nodes based on neighborhood distributions and prototype misalignment, while ACDM dynamically adjusts the distillation temperature to match the current difficulty level. This synergistic interaction enables more effective knowledge transfer of complex topological information.
>
>
> > `Question 1`: Implementation details of baselines?
>
> For FL methods, all baselines except FedAvg support model heterogeneity, and they work the same way as they do in computer vision tasks. But for FGL methods, to the best of our knowledge, we are the first to explore model heterogenity in FGL settings. Consequently, existing FGL methods (and also FedAvg) are originally designed for homogeneous models. To adapt them for our heterogeneous setting, we remove direct parameter sharing between architecturally different models. For evaluation, we distill a global model from local private models and assess its accuracy.
>
>
> > `Question 2`: Reasons for competition of temperature with model parameters?
>
> We are grateful that you asked this question, because we have the opportunity to explain to you once again an important feature of TRUST. When the proxy model learns well (low L), $\theta_{temp}$ increases to raise $\tau$, softening the possibility distributions and making the task harder to prevent underutilization of model capacity. When the model struggles (high L), $\theta_{temp}$ decreases to lower $\tau$, sharpening the task focus on high-confidence knowledge. In this way, $\tau$ self-adjusts to client capabilities without manual tuning.
>
>
> > `Formatting Concerns`
>
> We will correct the relevant typo in the revised version.

---

> > ### Comment · Reviewer_hwGA · 2025-08-05
> >
> > Thank you for the rebuttal. The authors' responses have addressed my main concerns, and I'll keep my positive score.

---

> > > ### Author Response · Authors · 2025-08-05
> > >
> > > Dear Reviewer hwGA,
> > >
> > > Thank you for supporting the acceptance of our work! We greatly appreciate your thoughtful review, and your insights have been valuable in improving our paper.
> > >
> > > Best regards,
> > >
> > > Authors

---

### Note · Authors · 2025-08-14

***Dear Area Chair and Reviewers,***

We sincerely thank the area chair for your exceptional leadership throughout this review process, and all reviewers for your insightful feedback. Your time and expertise have been invaluable in improving our work, and the careful consideration given to our research is deeply appreciated.

Regarding the rebuttal process, we are pleased to report that the majority of participating reviewers have acknowledged our comprehensive responses to their concerns, and have accordingly **voted in favor of accepting our manuscript**:

- `Reviewer hwGA` expressed satisfaction with our detailed methodology explanations and supplementary evaluation experiments examining hyperparameter sensitivity and ablation studies.

- `Reviewer SdVL` retained their positive evaluation following our additional comparisons with more FGL baselines.

- `Reviewer 6U5h` indicated approval of our expanded discussion of experimental details and supplementary evaluations.

- `Reviewer i4hd` appreciated our additional experiments where TRUST outperforms other FGL baselines across various evaluation metrics.

In response to the concerns raised by `Reviewer BwgV`, we applied the same level of careful attention and thoroughness. *First*, we offered supplementary experiments and theoretical justifications repsonding to each concern raised. *Additionally*, we provided a detailed follow-up response specifically addressing their remaining questions about the choices of baselines, significance of results and overall computing complexity.

However, we remain somewhat uncertain whether our revisions have fully resolved all of their concerns. We hope that the revisions we have made will be found satisfactory.

Overall, we believe the review process has provided a valuable opportunity to strengthen our work and address the reviewers' insightful comments. While we are pleased to have resolved the majority of concerns raised, we remain hopeful that the efforts we’ve made to refine our work will be thoroughly considered in your final evaluation.

Once again, we extend our sincere appreciation for your guidance and thoughtful consideration of our manuscript. We greatly value the AC's leadership and look forward to your fair and balanced evaluation of our work.

Best regards,

Authors

---

### Decision · Program_Chairs · 2025-09-17

**Decision:**

Accept (poster)

**Comment:**

The paper introduces TRUST, a novel framework for model-heterogeneous federated graph learning. The authors assert that they have addressed several key issues found in previous approaches in this area. The paper received five reviews, with high scores of 5, 5, 4, 4, and 3. Overall, the reviewers were impressed with the paper and supported its acceptance.